# Cyclic peptides discriminate BCL-2 and its clinical mutants from BCL-X$_L$ by engaging a single-residue discrepancy

Fengwei Li [1,7] ✉, Junjie Liu[2,7], Chao Liu[1,7], Ziyan Liu[2], Xiangda Peng[3], Yinyue Huang[1], Xiaoyu Chen[1], Xiangnan Sun[1], Sen Wang[1], Wei Chen[4], Dan Xiong[5], Xiaotong Diao [1], Sheng Wang[3], Jingjing Zhuang[1,6], Chuanliu Wu [2] ✉ & Dalei Wu [1] ✉

Overexpressed pro-survival B-cell lymphoma-2 (BCL-2) family proteins BCL-2 and BCL-X$_L$ can render tumor cells malignant. Leukemia drug venetoclax is currently the only approved selective BCL-2 inhibitor. However, its application has led to an emergence of resistant mutations, calling for drugs with an innovative mechanism of action. Herein we present cyclic peptides (CPs) with nanomolar-level binding affinities to BCL-2 or BCL-X$_L$, and further reveal the structural and functional mechanisms of how these CPs target two proteins in a fashion that is remarkably different from traditional small-molecule inhibitors. In addition, these CPs can bind to the venetoclax-resistant clinical BCL-2 mutants with similar affinities as to the wild-type protein. Furthermore, we identify a single-residue discrepancy between BCL-2 D111 and BCL-X$_L$ A104 as a molecular "switch" that can differently engage CPs. Our study suggests that CPs may inhibit BCL-2 or BCL-X$_L$ by delicately modulating protein-protein interactions, potentially benefiting the development of next-generation therapeutics.

Programmed cell death (apoptosis) occurs during the course of various important physiological processes, and its dysregulation is related to many diseases, such as cancer, autoimmune diseases and neurodegenerative disorders[1]. The B-cell lymphoma-2 (BCL-2) family proteins play a central role in the regulation of apoptotic cell death induced by a variety of stimuli[2]. The apoptosis "effectors" within this family are represented by BAX and BAK, and their activation triggers mitochondrial outer membrane permeabilization, committing cells to apoptosis through the mitochondrial pathway[3,4]. The anti-apoptotic (pro-survival) proteins of the BCL-2 family include BCL-2, BCL-X$_L$, MCL-1, BCL-w and BFL-1/A1. These proteins bind to and sequester the

apoptosis effectors, as well as the apoptosis sensitizers, BIM, BID, PUMA, NOXA, BAD, BMF, HRK and BIK[5,6]. Apoptosis effectors and anti-apoptotic proteins share four homologous domains referred to as the BCL-2 homology 1, 2, 3 and 4 (BH1, BH2, BH3 and BH4) domains. In contrast, the eight apoptosis sensitizers contain only the BH3 domain and therefore, are also called "BH3-only" proteins[3,4]. Cancer cells can undermine normal apoptosis by altering the balance among these apoptotic and anti-apoptotic proteins to gain a survival advantage[7,8]. Targeting the anti-apoptotic BCL-2-like proteins for cancer therapy is thus a very attractive strategy, since their overactivity promotes cancer cell growth and often limits tumor responses to cytotoxic agents[9].

[1]Helmholtz International Lab, State Key Laboratory of Microbial Technology, Shandong University, Qingdao 266237, China. [2]The MOE Key Laboratory of Spectrochemical Analysis and Instrumentation, State Key Laboratory of Physical Chemistry of Solid Surfaces, Department of Chemistry, College of Chemistry and Chemical Engineering, Xiamen University, Xiamen 361005, China. [3]Shanghai Zelixir Biotech Company Ltd., Shanghai 200030, China. [4]Shanghai Immune Therapy Institute, Shanghai Jiao Tong University School of Medicine-Affiliated Renji Hospital, Shanghai 200127, China. [5]Xiamen Lifeint Technology Company Ltd., Xiamen 361005, China. [6]Marine College, Shandong University, Weihai 264209, China. [7]These authors contributed equally: Fengwei Li, Junjie Liu, Chao Liu. ✉e-mail: lifengwei@sdu.edu.cn; chlwu@xmu.edu.cn; dlwu@sdu.edu.cn

BCL-2 is overexpressed in lymphoid malignancies, and it is the predominant survival factor in many of the related tumor types[10]. Meanwhile overexpression of BCL-X_L has been found correlated with drug resistance and disease progression of multiple solid tumors and hematologic malignancies[11,12]. Three-dimensional structural studies of BCL-2 and BCL-X_L each in complex with different BH3-only proteins[13–17] have provided valuable insights into how these two proteins interact with their pro-apoptotic counterparts. They have a very high structural similarity in consisting of eight to nine α-helices, with two mostly hydrophobic α-helices forming a structural backbone that is surrounded by the rest six to seven amphipathic α-helices[18–20]. An elongated hydrophobic groove is thus formed along the surface of BCL-2 and BCL-X_L proteins, spanning approximately 20 Å to serve as the binding site for the amphipathic α-helical BH3 domains of their pro-apoptotic partners[21]. This critical protein-protein interaction (PPI) brings a unique opportunity to discover inhibitors that bind to the surface groove of BCL-2 or BCL-X_L and block its interaction with the BH3 domains, though this is a relatively challenging task due to the large and flat shape of this groove.

Combining multiple approaches of structure-based drug design (SBDD), nuclear magnetic resonance (NMR)-based compound screening and parallel synthesis, a team from AbbVie, Inc. discovered ABT-737, a BH3-mimetic small-molecule inhibitor binding to the anti-apoptotic proteins BCL-2, BCL-X_L and BCL-w with a high affinity[22]. In order to improve the oral utilization, they later developed another small-molecule inhibitor navitoclax (ABT-263), exhibiting a 20% to 50% oral bioavailability in preclinical animal models[23]. However, because navitoclax binds not only to BCL-2 but also to BCL-X_L, this drug causes predictable, dose-dependent thrombocytopenia[24]. We need to point out that due to the high degree of structural similarity between the BH3-binding regions of BCL-2 and BCL-X_L, it is actually very difficult to design de novo BCL-2-selective inhibitors[8,18,22].

The AbbVie team exploited co-crystal structures of BCL-2 and small-molecule inhibitors to guide their rational design of venetoclax (ABT-199), a first-in-class, highly specific inhibitor of BCL-2, and one of the first approved small-molecule tumor therapeutics that directly blocks a PPI[25–28]. However, recent studies have found that after 19-42 months of venetoclax treatment, an induced G101V mutation in BCL-2 could be identified in a large portion of the patients[29–31]. G101V reduces the binding affinity of BCL-2 to venetoclax by ~180-fold as measured by surface plasmon resonance (SPR) assays[31]. The molecular mechanism of this drug-resistant mutation has been elucidated by co-crystal structures that the bulker side chain of valine would lead to the conformational change at a key position within the venetoclax binding pocket[31]. Another clinical mutation D103Y of BCL-2 occurring with a lower frequency has also been found in recent studies[27,32]. Therefore, there is an urgent need to develop the second-generation BCL-2-specific inhibitors that can bind both BCL-2 and its clinical mutants.

Peptides have been an emerging frontier for drug discovery in recent years, occupying a unique chemical space between small molecules and biologics to potentially integrate the merits of these two different drug modalities[33–35]. Targeting the PPIs between BCL-2 or BCL-X_L and their pro-apoptotic counterparts, researchers have successfully designed a series of modified short peptides, including the α/β-peptides[36,37] and stapled peptides[38–42]. These peptides exhibit increased helicity and decreased susceptibility to proteases, as compared with the conventional BH3-only peptides[37]. However, these modified peptides usually lack the binding selectivity to BCL-2 or BCL-X_L over the other BCL-2 family proteins. Compared with linear peptides, cyclic peptides (CPs) are more stable both in structure and against proteolysis, representing a promising class of ligands or therapeutics for those challenging drug targets including PPIs[34]. Many CPs binding to their targets with a high affinity and specificity have been developed in recent years[33]. These peptides can often benefit from their unique chiral and structural complexity to exert an exquisite

protein-binding potency[43]. In practice, CPs with diverse binding modes to their targets can be discovered through a high-throughput screening of CP libraries, although only a few complex structures of target proteins and bound CPs have been reported[43].

In this study, by combining both co-crystal structural analysis and molecular dynamics (MD) simulations, we revealed the structural mechanism of CPs targeting BCL-2 or BCL-X_L proteins, especially how their binding mode is different from those of small-molecule inhibitors. Then we obtained CPs with improved binding affinity or altered selectivity to BCL-2 or BCL-X_L, and identified a pair of non-conserved residues (BCL-2 D111 and BCL-X_L A104) that influence the binding of CPs. Our study may provide insights for the development of next-generation BCL-2/BCL-X_L inhibitors that can potentially bypass drug-resistant clinical mutants.

## Results

### Structural basis for the selectivity of cp1 to BCL-2

In our previous work[44], a BCL-2-targeting screening against a phage-based 2-aryl-4,5-dihydrothiazole (ADT)-cyclic peptide library enabled the identification of cADT-CB-5 (annotated as cp1 hereafter and its chemical structure shown in Fig. 1a, Supplementary Table 1, Supplementary Fig. 1). Since the overall structures and surface grooves of BCL-2 and BCL-X_L proteins are highly similar as mentioned above, we wanted to know whether cp1 can also bind to BCL-X_L and whether it exhibits a preference between two proteins. Therefore, we measured the binding affinities of cp1 to both BCL-2 and BCL-X_L using a SPR assay. The results indicate that cp1 has a much higher binding affinity to BCL-2 ($0.65 \pm 0.14$ µM) than to BCL-X_L (>12 µM) (Table 1), implying an over 19-fold selectivity of cp1 for these two highly similar BCL-2 family proteins.

To understand the beneath structural basis for this selectivity of cp1, we co-crystalized and determined the complex structures of cp1 bound to BCL-2 and BCL-X_L proteins with high resolutions at 2.1 Å and 2.0 Å, respectively. Detailed crystallographic data statistics are shown in Supplementary Table 2. In both structures, the electron densities for cp1 were well defined (Supplementary Fig. 2). It is not very surprising that like the small-molecule inhibitors and BH3-mimetic peptides, cp1 also binds to the hydrophobic surfaces (formed by α2, α4, α5, α8 helices) of BCL-2 or BCL-X_L where the BH3-only proteins usually bind (Fig. 1b, c). Interestingly, the conformation of cp1 is almost identical in these two different complexes (Fig. 1d). In addition, the overall conformations of BCL-2 and BCL-X_L bound by cp1 are also very similar, as indicated by the RMSD of only 0.26 Å for all their Cα atoms (Fig. 1d). These results suggest that cp1 may interact with BCL-2 and BCL-X_L in a highly similar mode.

Taking a closer look, we analyzed in detail the key amino-acid residues within 5 Å around cp1 in two complex structures to seek for differences, and found that most of these residues are conserved in BCL-2 and BCL-X_L, except for two pairs of residues (i.e. D111, M115 in BCL-2 and corresponding A104, L108 in BCL-X_L) (Fig. 1e). Two residues of cp1 (i.e. cp1-Y5 and cp1-W7) are buried in the hydrophobic pocket formed by F104, Y108, F112, M115, L137, F138 and A149 of BCL-2 (F97, Y101, F105, L108, L130, F131 and A142 of BCL-X_L). The cp1-R4 forms hydrogen bonds with E136 and L137 of BCL-2 (E129 and L130 of BCL-X_L), and cp1-D8 forms a salt bridge with R146 of BCL-2 (R139 of BCL-X_L) (Fig. 1e–g, Supplementary Fig. 3a, b), a conserved arginine residue of the pro-survival proteins known to interact with a conserved aspartate residue of the BH3-only proteins[45] (Supplementary Fig. 3c, d). It is noteworthy that the main-chain N atom of cp1-G6 forms a hydrogen bond with the carboxyl group of D111 in BCL-2 (Fig. 1f). However, in BCL-X_L, the corresponding residue is A104, which has a shorter side chain and can't directly interact with cp1-G6 (Fig. 1g). Another pair of different residues, M115 of BCL-2 and L108 of BCL-X_L, are both hydrophobic amino acids, and they are located at bottom of the big hydrophobic pocket, with their minimum distances to cp1 measured as

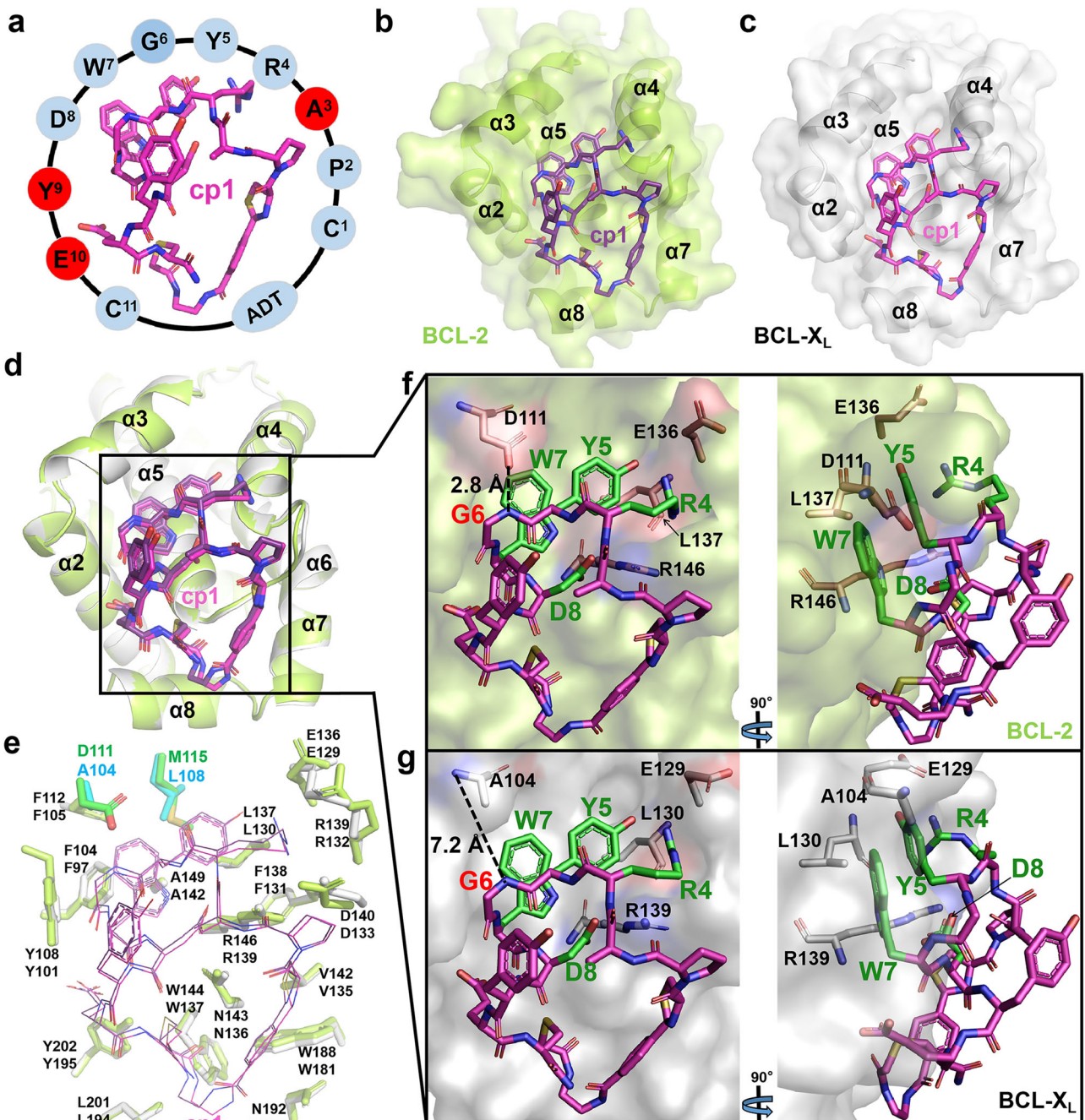

**Fig. 1 | Structural basis for the selectivity of cp1 to BCL-2. a** 3D structure model of cp1 with its residue sequence shown around the cycle. The residues not directly involved in the cp1-protein interactions are shown in red. The overall binding positions of cp1 upon BCL-2 (**b**) and BCL-X$_L$ (**c**). BCL-2 and BCL-X$_L$ proteins are shown as surface in limon (**b**) or white (**c**), with their α-helices near the binding groove labeled; and cp1 is shown as sticks in purple (**b**) or magenta (**c**). **d** Superposition of BCL-2 and BCL-X$_L$ in complex with cp1. Colors are the same as in panels b and c. All 7 α-helices of two proteins surrounding cp1 are labeled accordingly. **e** Comparison of key amino-acid residues within 5 Å around cp1 in BCL-2 and BCL-X$_L$. Two pairs of non-conserved residues, D111/M115 of BCL-2 and A104/L108 of BCL-X$_L$ are shown as sticks in green and cyan, respectively. Key residues of cp1 interacting with BCL-2 (**f**) or BCL-X$_L$ (**g**) in two different views.

4.3 Å and 4.4 Å respectively (Supplementary Fig. 4). Therefore, we speculate that the D111/A104 discrepancy between BCL-2 and BCL-X$_L$ may be mainly responsible for the selectivity of cp1, while the M115/L108 pair is perhaps not as crucial as the former one.

To test the essentiality of D111, we generated a BCL-2 D111A mutant and measured its binding affinity to cp1 also using SPR. This mutant showed a clearly lower cp1-binding affinity (5.34 ± 1.00 μM) than wild-type (WT) BCL-2 (0.65 ± 0.14 μM), yet comparable to BCL-X$_L$ (>12 μM) (Table 1). These results suggest that the D111 residue of BCL-2 plays a key role in the preferential binding of cp1 to BCL-2 over BCL-X$_L$.

**A unique binding mode of cp1 upon BCL-2 family proteins**

After solving the crystal structures of BCL-2 and BCL-X$_L$ each in complex with cp1, we wanted to compare the binding mode of CP with those of previously reported ligands upon two proteins. Since the first crystal structure of BCL-X$_L$ (PDB ID: 1MAZ)[19] solved in 1996, a large number of crystal structures of BCL-2 and BCL-X$_L$ have been reported. According to the nature of their ligands, previous BCL-2 and BCL-X$_L$ complex structures can be roughly divided into three groups, respectively bound by small-molecular inhibitors, BH3-only peptides and BH3-mimetic peptides.

**Table 1 | Binding affinities of BCL-2 family proteins and BCL-2 mutants to different CPs and S55746**

| | cp1 $K_D$ (µM) | cp2 $K_D$ (µM) | cp3 $K_D$ (µM) | cp4 $K_D$ (µM) | cp5 $K_D$ (µM) | S55746 $K_D$ (µM) |
|---|---|---|---|---|---|---|
| BCL-2 | 0.65 ± 0.14 | 0.20 ± 0.01 | 0.24 ± 0.02 | 0.08 ± 0.01 | 0.08 ± 0.01 | 0.006 ± 0.002 |
| BCL-X$_L$ | >12.14 | 0.73 ± 0.18 | 0.09 ± 0.01 | 0.47 ± 0.05 | 0.04 ± 0.01 | 0.175 ± 0.042 |
| BCL-2-D111A | 5.34 ± 1.00 | 0.12 ± 0.01 | 0.09 ± 0.02 | | | 0.006 ± 0.003 |
| BCL-2-G101V | 1.00 ± 0.13 | 0.30 ± 0.07 | 0.66 ± 0.004 | | | |
| BCL-2-D103Y | 0.36 ± 0.03 | 0.21 ± 0.05 | 0.34 ± 0.04 | | | |
| BCL-2-F104L | 1.62 ± 0.36 | 0.26 ± 0.02 | 0.71 ± 0.07 | | | |
| BCL-w | 1.98 ± 0.39 | 1.45 ± 0.32 | 0.22 ± 0.04 | | | |
| MCL-1 | NB | NB | NB | | | |
| BFL-1 | NB | NB | NB | | | |

These results were determined by SPR assays using the affinity fitting method, and indicated by mean values from three independent experiments with the standard deviation (SD).
$K_D$ dissociation constant, *NB* no binding.

Although there is still no crystal structure of ligand-free BCL-2 available, its α3 and α4 helices would together deflect 3–5° counterclockwise when bound by small-molecule inhibitors or BH3-only peptides, as compared with the NMR structure of unbound BCL-2 (Fig. 2a). As previously reported[31,45,46], induced conformational changes of the α3 and α4 helices or just a partial unfolding of the helix α3 are critical for BCL-2 family proteins to identify different BH3-only proteins, and also very important for effective targeting of traditional small-molecule inhibitors. For example, BAX induces the conformational changes of α3 and α4, and its four residues L59, L63, I66 and L70 respectively binds into P1, P2, P3 and P4 pockets of BCL-2, with the residue D68 forming a salt bridge with BCL-2 R146 (Fig. 2b). Similarly, venetoclax induces conformational changes of these two helices to expose the deep ligand-binding pockets P2 and P4 to accommodate its binding (Fig. 2c). For BCL-X$_L$, crystal structures indicate that its α3 and α4 helices also deflect 5–10° counterclockwise upon the binding of BH3-only peptides or traditional inhibitors, compared with the unbound state (Fig. 2e). As shown in Fig. 2f, g, the binding of BH3-only protein BID and BH3-mimetic inhibitor ABT-737 to BCL-X$_L$, also needs to induce conformational changes of α3 and α4 helices to fit into corresponding pockets.

However, upon cp1 binding, the overall conformation of α3 and α4 helices of BCL-2/BCL-X$_L$ remains roughly the same as in the crystal structure of ligand-free BCL-X$_L$, still exhibiting a flat interaction surface similar to the unbound state (Fig. 2a, d, e, h). To further assess the special binding mode of cp1, we quantitatively measured and compared the solvent-accessible surface area (SASA) values of certain key residues on the binding surface after engaged by different ligands/binders (Supplementary Fig. 5). The respective SASAs in the structures of BCL-2-BAX, BCL-2-venetoclax, BCL-2-cp1 are 979 Å², 1007 Å², 768 Å², while in those of BCL-X$_L$-BID, BCL-X$_L$-ABT-737, BCL-X$_L$-cp1 are 1135 Å², 1061 Å², 828 Å² (as illustrated in Supplementary Fig. 6), implying relatively smaller SASA values upon cp1 binding for both BCL-2 and BCL-X$_L$. These results suggest that in contrast to previously reported small-molecule or BH3-related ligands, cp1-like CPs do not induce conformational changes of BCL-2/BCL-X$_L$ to unveil their deep pockets (such as P2 and P4) upon binding. Therefore, through a comparative structural analysis we found that cp1 targets BCL-2/BCL-X$_L$ in a binding mode that does not need to induce conformational changes of α3 and α4 helices, which is clearly different from the other inhibitors with their complex structures reported previously (Supplementary Fig. 5).

**Cp1 also binds to venetoclax-resistant BCL-2 mutants**

As the long-term clinical application of venetoclax has caused an occurrence of drug-resistant BCL-2 mutations (such as G101V and D103Y), it is critical to seek for inhibitors potentially overcoming them and figure out the mechanism of action. A recent study revealed that the G101V mutation induces subtle conformational change of the P2

pocket in BCL-2 (Fig. 3a), which subsequently reduces the binding and function of selective inhibitors[31], such as venetoclax and S55746. Since cp1 shows a unique binding mode to BCL-2 and does not insert into the P2 pocket (Fig. 2), we speculate its binding may not be affected by the clinical mutation G101V.

To examine this hypothesis, we obtained the crystals of BCL-2 G101V mutant in complex with cp1 that diffracted to a high resolution (1.85 Å) in space group $P2_1$ (Supplementary Table 2). The overall structure of BCL-2 G101V is similar to WT, except for the N-terminus of α4 helix, which is relatively far away from the cp1 binding site (Fig. 3b, Supplementary Fig. 7a, b), and is likely not directly involved in the binding of cp1 to G101V mutant. The electron density of cp1 in BCL-2 G101V complex was also well defined (Supplementary Fig. 2), showing a highly similar conformation for cp1 as in the WT structure (Fig. 3b). In the G101V-cp1 complex structure, the E152 residue on α5 helix of BCL-2 shows a 60° rotamer change relative to its counterpart in the WT protein, same as in the structures of G101V-venetoclax and G101V-S55746 complexes reported previously[31] (Fig. 3c–g). However, the minimum distance between E152 and cp1 is about 7.7 Å, much further than those between E152 and venetoclax or S55746 in their complexes with BCL-2 G101V mutant, 3.3 Å and 3.9 Å, respectively (Fig. 3d–f). This distance difference suggests that the knock-on effect of G101V mutation on the adjacent E152, which accounts for its drug resistance to venetoclax or S55746 as previously illustrated[31], may not drastically impact the binding of cp1 to the BCL-2 G101V mutant. To confirm this, an SPR assay was conducted and the results showed that the affinity of cp1 to BCL-2 G101V mutant was 1.00 ± 0.13 µM (close to the 0.65 ± 0.14 µM for BCL-2 WT), indicating no drastic influence on cp1 binding (Table 1).

Interestingly, in the BCL-2 G101V-cp1 complex we found the conformational change of E152 residue may further lead to a 90° rotamer change of neighboring F112, as compared with its counterpart in the WT-cp1 complex (Fig. 3c, d). Probably as an accommodated move, the nearby F104 residue on α2 also shows a 90° rotamer change, which is not observed in the complex structures of BCL-2 WT-cp1, G101V-venetoclax and G101V-S55746 (Fig. 3d–f). These results indicate that the BCL-2 mutation G101V may not only influence the conformation of E152 as previously reported[31], but also cause conformational changes of F112 and F104 (Fig. 3d–f). One possible reason why in the BCL-2 G101V-venetoclax or G101V-S55746 complex structures, these two residues are not showing knock-on effects as E152 by the G101V mutation could be that their accommodated conformational changes are restrained by the binding of small-molecule drugs, which is quite different from the cp1-binding scenario. More importantly, F112 and F104 are located in the P2 and P3 pockets, respectively (Fig. 3a), thus potentially participating in the interactions between BCL-2 and small-molecule inhibitors. These results suggest that the accompanied changes of nearby residues F112 and F104 may also influence the

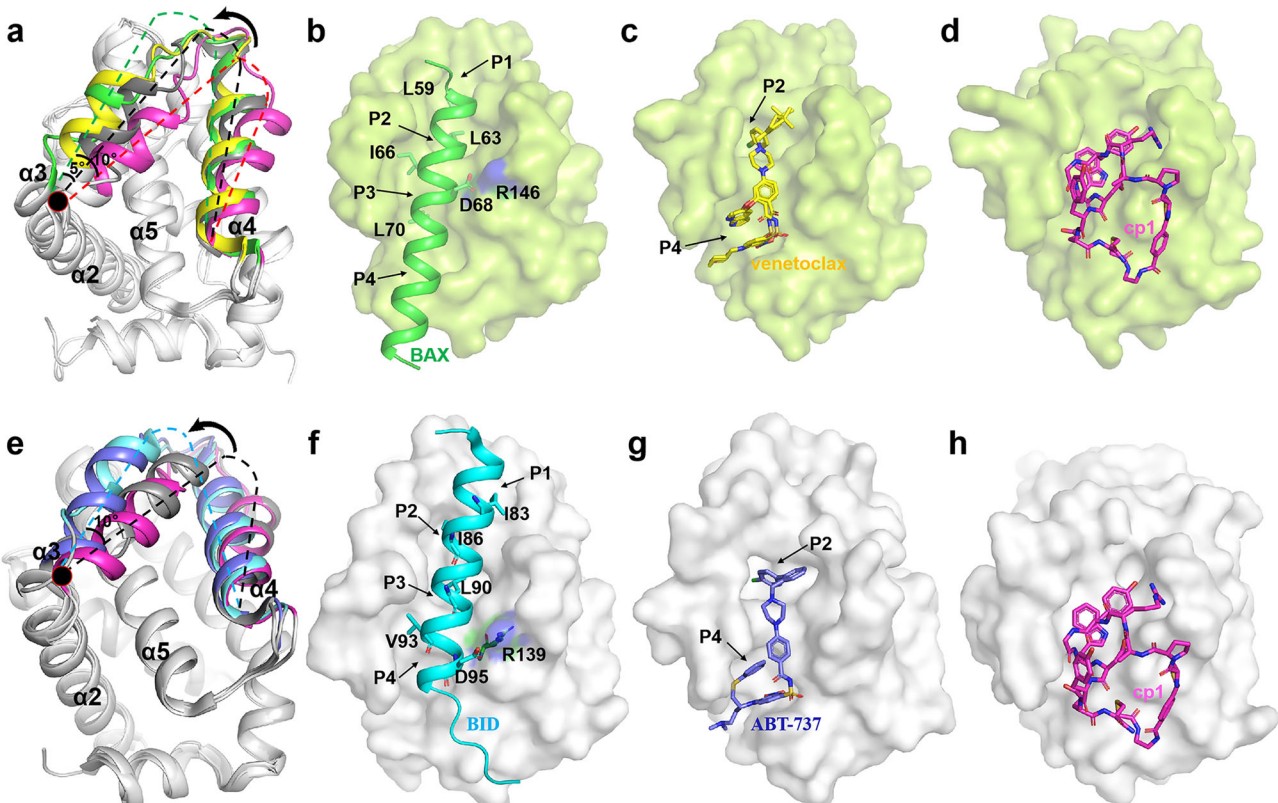

**Fig. 2 | A special binding mode of cp1 targeting BCL-2 and BCL-X$_L$ proteins. a** Conformational comparison of BCL-2 bound by different ligands. The overall structure of BCL-2 is shown as cartoon. The α3 and α4 helices (F112-R139) from superposed structures of BCL-2-cp1 complex, BCL-2-BAX complex (PDB ID: 2XA0), BCL-2-venatoclax complex (PDB ID: 6OOK) and ligand-free NMR structure of BCL-2 (PDB ID: 1GJH) are colored in magenta, green, yellow and gray, respectively. The spatial locations of these two helices are marked by dashed lines to indicate their conformational changes. **b** The conformation of BAX (green) in complex with BCL-2. The BAX residues L59, L63, I66 and L70 are shown as sticks binding into the P1, P2, P3 and P4 pockets (marked with black arrows), respectively. **c** The conformation of venetoclax (yellow) in complex with BCL-2. **d** The conformation of cp1 (magenta) in complex with BCL-2. **e** Conformational comparison of BCL-X$_L$ bound by different ligands. The overall structure of BCL-X$_L$ is shown as cartoon. The α3 and α4 helices (F105-R132) from superposed crystal structures of ligand-free BCL-X$_L$ (PDB ID: 1AF3), BCL-X$_L$-cp1 complex, BCL-X$_L$-ABT-737 complex (PDB ID: 2YXJ) and BCL-X$_L$-BID complex (PDB ID: 4QVE), are colored in gray, magenta, light blue and cyan, respectively. **f** The conformation of BID (cyan) in complex with BCL-X$_L$. The BID residues I83, I86, L90 and V93 are shown as sticks binding into P1, P2, P3 and P4 pockets, respectively. **g** The conformation of ABT-737 (light blue) in complex with BCL-X$_L$. **h** The conformation of cp1 (magenta) in complex with BCL-X$_L$.

binding affinities of venetoclax and S55746 to BCL-2 mutant G101V. We noticed that a previous study using a related mouse model identified drug-resistant mutations BCL-2 F104L and F104C (corresponding residue numbers in the human protein), which could strongly decrease venetoclax binding[29,31,47]. However, the binding affinity of cp1 to BCL-2 F104L mutant was measured as $1.62 \pm 0.36\,\mu M$ by SPR (Table 1), only showing a relatively small affinity reduction and again highlighting the unique binding mode of this CP.

**Molecular mechanism of venetoclax-resistant G101V mutation**

To better clarify the mechanism of drug resistance of clinical BCL-2 mutations and why cp1 binding is not severely compromised by these mutations, we used a MD simulation method to test the chi angle changes of certain key residues of BCL-2 at its well-defined interface to the BH3-only proteins (Fig. 3a). The chi angle is one of the most important features of residue side-chain conformation, and it is widely used for the conformational analysis or the description of interactions involving side chains[48,49] (Supplementary Fig. 7c). In the simulated ligand-free state of both BCL-2 WT and G101V, the chi1 angles of F104 and F112 can distribute in either the -1.5-radian Peak1 or the 3-radian Peak2 (Fig. 3h). When cp1 binds to them in the simulation, the chi1 angles of F104 and F112 could still distribute in both peaks, although with a tendency leaning to the Peak2 as compared with the ligand-free state (Fig. 3i). Meanwhile in the cp1-bound complex crystal structures, the chi1 angles of F104 and F112 are shown both in Peak2 for BCL-2 WT

or both in Peak1 for G101V (Fig. 3i). These results imply that cp1 binding would not dictate the chi1 angle distribution of these key residues in BCL-2 WT or G101V proteins.

In contrast, MD simulation suggested that upon venetoclax binding to BCL-2 WT or G101V, the distribution of chi1 angles of F104 and F112 would both shift to the 3-radian Peak2 (Fig. 3j). In previously reported crystal structures of BCL-2 WT and G101V each in complex with venetoclax[31], the chi1 angles of these two residues are also shown in Peak2, consistent with our MD simulation results (Fig. 3j). Then we conducted a simulation in the same way for the chi1 angel distribution of these two residues upon S55746 binding (Fig. 3k). Interestingly, MD results showed that for both BCL-2 WT and G101V, the binding of S55746 would dramatically concentrate the chi1 angle distribution of F104 in the Peak2 while that of F112 in the Peak1, which is consistent with their chi1 angles distribution in the BCL-2-S55746 crystal structure (Fig. 3k). These data correlate well with our aforementioned speculation that binding of small-molecule inhibitors may limit the conformational changes of F104 and F112 caused by G101V mutation.

In addition, MD simulation showed that venetoclax binding could also influence the chi angles of M115 and R139 (Fig. 3a, Supplementary Fig. 7d), in line with the known conformational changes of α3 and α4 helices (Fig. 2a). We used a correlation analysis to explore the possible linkage between side-chain conformational changes of certain key residues near the BH3-binding groove within 5 Å around cp1 (Supplementary Fig. 5). The results showed that G101V mutation directly

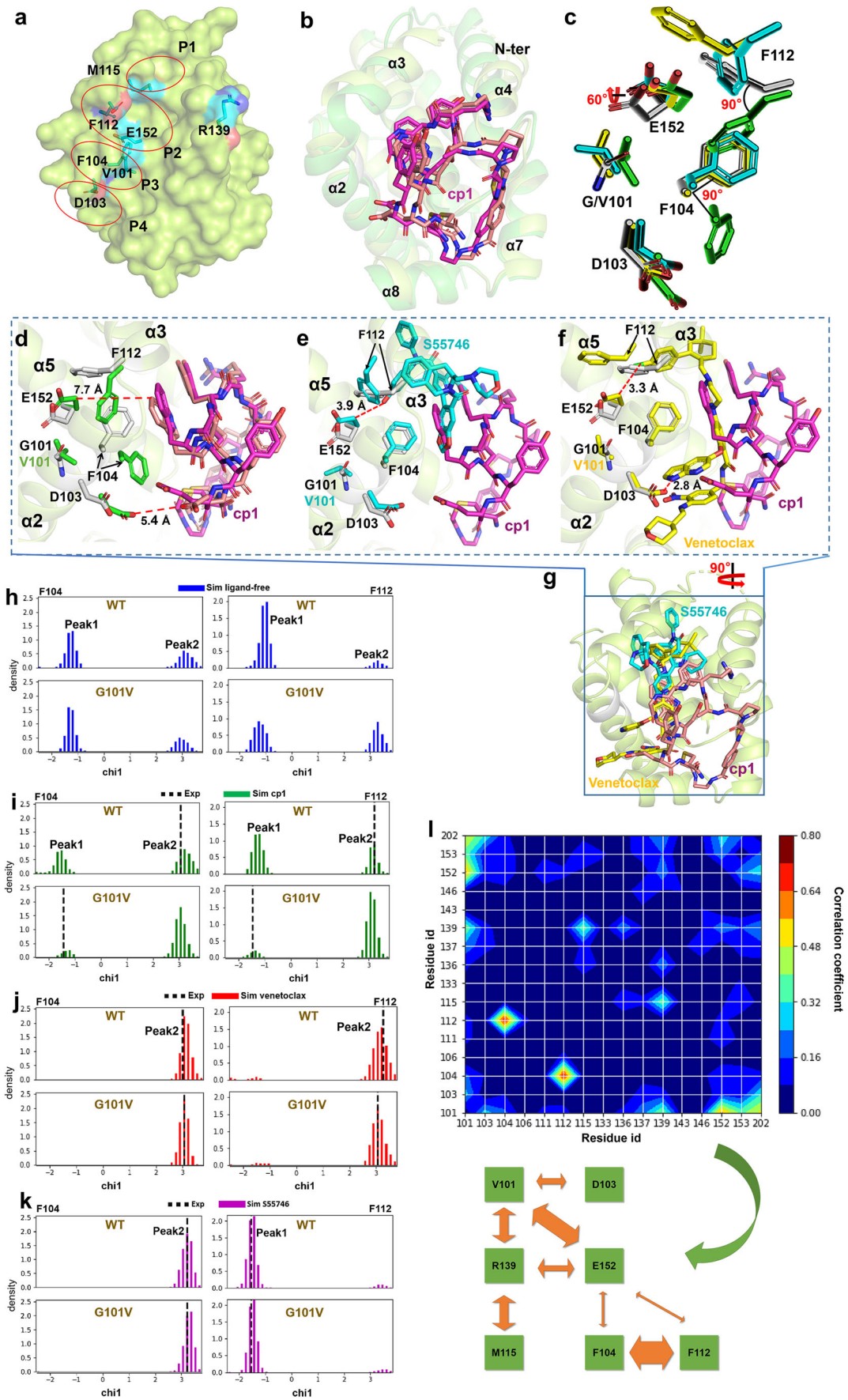

**Fig. 3 | The structural mechanism of cp1 against BCL-2 G101V mutation.**
**a** Locations of key residues in the BH3-binding groove of BCL-2. The four binding pockets (P1-P4) are marked by red ellipses. **b** Comparison of cp1-bound BCL-2 WT and BCL-2 G101V complexes. The overall structures of BCL-2 WT and G101V are shown as cartoon in limon and green, with the cp1 in complex with WT and G101V respectively colored in magenta and brown. **c** The conformation comparison of key BCL-2 residues G/V101, D103, F104, F112 and E152 in complexes with different ligands. The residues from BCL-2 WT-cp1, BCL-2 G101V-cp1, BCL-2 G101V-S55746 and BCL-2 G101V-venetoclax complexes are shown as sticks in gray, green, cyan and yellow, respectively. Conformation of key BCL-2 residues potentially involved in ligand-binding in the BCL-2 WT-cp1 complex (gray) as compared with those in the complexes of BCL-2 G101V-cp1 (**d**, green), BCL-2 G101V-S55746 (**e**, cyan) and BCL-2

G101V-venetoclax (**f**, yellow). **g** Comparison of the binding positions of cp1, venetoclax and S55746 upon BCL-2. The MD simulation results showing chi1 angle distributions of BCL-2 residues F104 (left) and F112 (right) from the WT or G101V mutant in the states of ligand-free (**h**, blue), cp1-bound (**i** green), venetoclax-bound (**j** red) and S55746-bound (**k** magenta), respectively. Solid lines represent simulation (Sim) distribution of chi1 angles, while dotted lines indicate their experimental (Exp) status in the crystal structures. **l** Correlation between conformational changes of different BCL-2 residues. For residue 101, the amino acid type (G/V) was used as the state in simulation, with 0 for G and 1 for V. For D103, the chi2 angle was used as its state. While for the other residues, the chi1 angle was chosen as the state indicator. The color from red to blue indicates the correlation from high to low, while thickness of the orange arrows represents the strength of their correlation.

affects the side-chain status of residues D103, R139 and E152 (Fig. 3l). Residue R139 in turn induces a twist of the chi1 angle of M115, while E152 could be a key bridge that affects the chi angle distribution of F104, F112 and R139. Interestingly, there is a strong correlation between residues F104 and F112 (Fig. 3l). The results above imply that these correlated and dynamic BCL-2 residues may interact together to form a network potentially affecting ligand binding, by their coordinated side-chain conformational changes.

## Structure-based design and phage-based screening for improved CPs

Next, we attempted to discover more CPs potentially possessing improved selectivity, binding affinity or activity. Analyzing the complex structures of BCL-2 and BCL-$X_L$ each bound with cp1, we found several cp1 residues (R4, Y5, W7 and D8) directly involved in and likely crucial for protein-CP interactions (Fig. 1f, g). Therefore, we constructed a special phage-based library by retaining these key residues (i.e. -CPXRYXWDXXC- with X standing for any amino-acid residue), and screened against both BCL-2 and BCL-$X_L$ proteins[50–52] (Fig. 4a). From the BCL-2 targeted screening we obtained a CP annotated as cp2, which exhibited enhanced binding affinities to both BCL-2 (0.20 ± 0.01 μM) and BCL-$X_L$ (0.73 ± 0.18 μM) as measured by SPR (Table 1, Fig. 4a). However, it appears cp2 is only 3.6-fold preferential for BCL-2 over BCL-$X_L$, much lower than cp1's >19-fold selectivity to BCL-2. On the other hand, we obtained cp3 from the BCL-$X_L$ targeted screening. The affinities of cp3 to BCL-$X_L$ and BCL-2 were measured as 0.09 ± 0.01 μM and 0.24 ± 0.02 μM, showing a clearly improved binding to BCL-$X_L$ as compared with cp1 (Table 1, Fig. 4a). We surprisingly noticed that only the CPs containing a proline residue at the 6th position could be enriched in our screening against BCL-$X_L$ (Supplementary Table 3), implying this residue may be essential in the binding and selectivity of CPs. To confirm the binding preferences of BCL-2 and BCL-$X_L$ towards these CPs, we adopted another binding assay using fluorescence polarization (Supplementary Fig. 8), which showed a similar tendency as the above SPR results suggested.

Despite their different performances in targeting BCL-2 and BCL-$X_L$ proteins, cp2 and cp3 actually share a very similar composition with only three residues not identical (E6, E9, F10 of cp2 *versus* P6, V9, E10 of cp3 as shown in Fig. 4a). To directly visualize the interactions between these two CPs and BCL-2 or BCL-$X_L$, we solved the high-resolution complex structures of BCL-2-12M-cp2, BCL-$X_L$-cp2, and BCL-$X_L$-cp3 (detailed crystallographic data statistics shown in Supplementary Table 2). The electron density maps of cp2 and cp3 were all clearly defined (Supplementary Fig. 2). Because it was very difficult to obtain crystals of BCL-2 WT proteins in complex with cp2, we adopted a surface-residue mutated version of BCL-2 protein, BCL-2-12M[13], which exhibits an overall three-dimensional structure as same as in the BCL-2-cp1 complex (Supplementary Fig. 9a).

## A single-residue "switch" affecting the selectivity of CPs

In order to figure out how cp2 exhibits a higher binding affinity than cp1, we compared and analyzed the binding conformations of cp1 and

cp2 and their surrounding key residues within 5 Å in the BCL-2 complexes (Supplementary Table 4). Notably, cp2 forms an additional electrostatic interaction with BCL-2 residue R110 through its E6 residue (Fig. 4b). Another clear conformational difference between cp1 and cp2 comes from the side-chain positions of cp1-E10 and cp2-F10. The BCL-2 residues around cp2-F10 are all hydrophobic ones (e.g. L201, Y202), which could be the reason why they trap the cp2-F10 in its downward position as compared with cp1-E10's upward position (Fig. 4b). The side chains of two newly introduced residues cp2-I3 and cp2-E9 (corresponding to Y3 and A9 of cp1) are facing outwards, and thus may not participate in the cp2-protein interactions directly (Fig. 4b). Therefore, cp2-E6 seems one of the possible key factors for the enhanced affinity of cp2 as compared with cp1. Interestingly, the corresponding G6 in cp1 is very important for its selectivity to BCL-2 over BCL-$X_L$. We found that BCL-2 D111 residue can still form a hydrogen bond with the main-chain N atom of cp2-E6, just like with the N atom of cp1-G6 (Figs. 1f and 4b). Meanwhile it is noteworthy that the side chain of cp2-E6 can also interact with BCL-2 R110 and BCL-$X_L$ R103 residues, respectively (Fig. 4b, c). Therefore, this special interacting mode of cp2-E6 may partially explain the enhanced binding affinities of cp2 to both BCL-2 and BCL-$X_L$, as well as its reduced selectivity to BCL-2 when compared with cp1 (Table 1).

To validate the above hypothesis, we synthesized cp4, a derivative of cp2 with the E6 substituted by G6 (Supplementary Figs. 1, 9c). Unexpectedly, the SPR data showed that the binding affinities of cp4 to BCL-2 and BCL-$X_L$ were 0.08 ± 0.01 μM and 0.47 ± 0.05 μM, suggesting both the binding preference and affinity of cp4 to BCL-2 are improved as compared with cp2 (Table 1). These results and aforementioned data of cp1 together imply that the 6th residue of CPs is likely one of the key positions affecting their binding affinity and selectivity to BCL-2/BCL-$X_L$, in correlation with the pair of non-conserved residues D111/A104 affecting the protein-CP interactions.

In addition to BCL-2, we next compared how BCL-$X_L$ differently interacts with these CPs. The overall three-dimensional structures of BCL-$X_L$ in complex with cp1, cp2 and cp3 are very similar (RMSD for all their Cα atoms between BCL-$X_L$-cp1 and BCL-$X_L$-cp2, BCL-$X_L$-cp1 and BCL-$X_L$-cp3, BCL-$X_L$-cp2 and BCL-$X_L$-cp3 are 0.216 Å, 0.199 Å and 0.171 Å, respectively) (Supplementary Fig. 9b). We again analyzed the key residues within 5 Å around these CPs, and found that there are more BCL-$X_L$ residues interacting with cp2 than cp1, in line with cp2's larger interaction area (the buried interfaces of cp1 and cp2 to BCL-$X_L$ are 592.6 Å² and 638.8 Å²) (Supplementary Table 4). This finding correlates with the higher binding affinity of cp2 to BCL-$X_L$ (0.73 ± 0.18 μM) than that of cp1 (Table 1). However, the buried interface between cp3 and BCL-$X_L$ is only 464.8 Å², although its binding affinity to BCL-$X_L$ is the highest (0.09 ± 0.01 μM) among the three CPs.

Subsequently, we further analyzed the interactions between cp3 and BCL-$X_L$, and found that cp3, like a small protein, can be roughly divided into three sites that each directly contact the corresponding sites of BCL-$X_L$ respectively (Fig. 4d–g). These three sites of cp3 exhibit different properties, with Site 1' (cp3-D8) negative charged, Site 2' formed by linker of CPs (cADT) and Site 3' composed by some

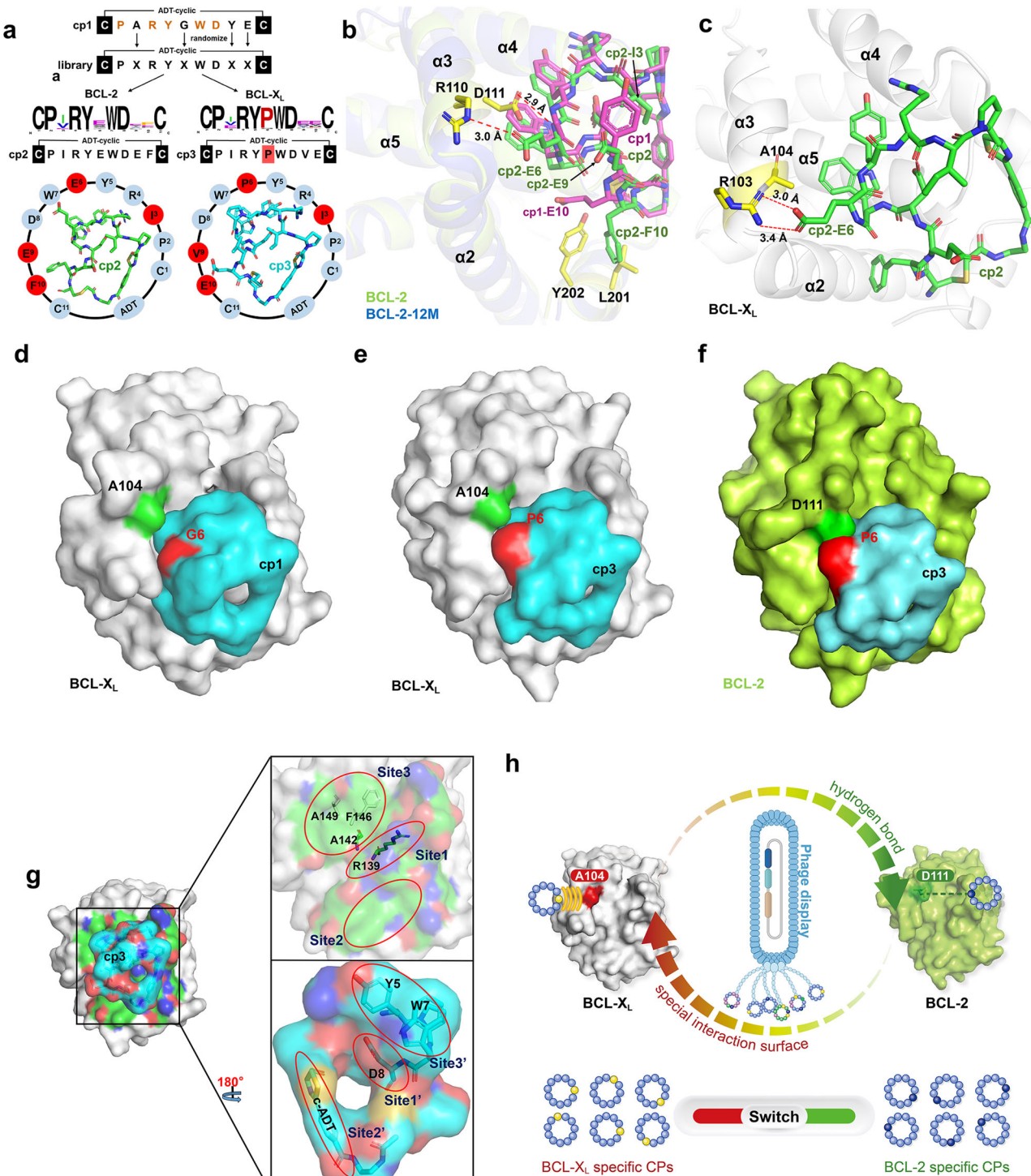

**Fig. 4 | Structure-based design and phage-based screening for improved CPs.** **a** The second round of phage display screening against BCL-2 and BCL-X$_L$ yielding cp2 (green) and cp3 (cyan), respectively. **b** The comparison between key binding sites of cp1 (magenta) and cp2 (green) in complex with BCL-2 (limon) and BCL-2-12M (light blue), respectively. Key residues of cp1 or cp2 binding shown as sticks in yellow. **c** Interactions between cp2 (green) and BCL-X$_L$, with the key residues highlighted in yellow. Structures in the "surface" view showing interactions between BCL-X$_L$ (**d**, **e**) or BCL-2 (**f**) and cp3 (**e**, **f**) or cp1 (**d**). The protein residues (A104 of BCL-X$_L$ or D111 of BCL-2) and CP's 6th residues (P6 or G6) are colored in green and red, respectively. **g** Dissected interface between BCL-X$_L$ (green) and cp3 (cyan). Special binding sites on BCL-X$_L$ and cp3 shown as sticks in green and cyan respectively. The key binding sites are labeled and marked by red ellipses. **h** Simplified mechanism of the special single-residue "switch" defining selectivity of CPs in phage-based screenings against BCL-X$_L$ or BCL-2.

hydrophobic residues (cp3-Y5, cp3-W7) (Fig. 4g). Among the 20 natural amino acids, proline is the only cyclic amino acid with a five-membered ring in the main chain and is often used to strengthen the rigidity of CPs[53]. The cp3-P6 forms a special molecular surface, which

can perfectly fit into the unique interface formed by A104 of BCL-X$_L$ (Fig. 4e). However, in the BCL-X$_L$-cp1 complex, there is some unfilled space between cp1-G6 and A104 of BCL-X$_L$, potentially causing a weaker interaction at this position (Fig. 4d). Notably, the

corresponding residue to BCL-X$_L$ A104 in BCL-2 is D111, which has a long side chain and would cause a steric clash with the surface of cp3 at P6 (Fig. 4f).

To test this hypothesis, we measured the binding affinity of BCL-2 D111A to cp3, and obtained a $K_D$ value of 0.09 ± 0.02 μM (comparable to the $K_D$ value of BCL-X$_L$ to cp3), indicating a nearly 3-fold improved affinity than that of BCL-2 WT (0.24 ± 0.02 μM) (Table 1). To further validate the key role of 6th residue of CPs, we synthesized cp5, a derivative of cp2 with E6 substituted by P6 (Supplementary Fig. 9d). As expected, cp5 showed higher binding affinities both to BCL-X$_L$ (0.04 ± 0.01 μM) and to BCL-2 (0.08 ± 0.01 μM) (Table 1). This proline substitution aiming at its specific interaction with BCL-X$_L$ A104 increased the binding affinity about 18-fold than cp2, and overturned the preference from BCL-2 to BCL-X$_L$ (Table 1). The above results suggest that this region of the protein-CP interface may be very important for binding affinity and selectivity.

In summary, we obtained several CPs with improved binding profiles targeting BCL-2 or BCL-X$_L$ proteins by rational design-based screening and modification. Moreover, in this process we identified and verified a single-residue discrepancy between BCL-2 D111 and BCL-X$_L$ A104 located at a key position of the surface BH3-binding groove, which could potentially serve as a molecular "switch" for the future discovery of highly selective inhibitors against BCL-2 or BCL-X$_L$ (Fig. 4h).

## The functional effects of CPs on tumor cells

After analyzing the direct interactions between CPs and BCL-2 or BCL-X$_L$ proteins by structural and biochemical approaches, we were curious about their functional effects within cells, especially whether CPs could inhibit a wide panel of leukemia and solid tumor cells like small-molecule inhibitors. At beginning, we used venetoclax as a positive compound and screened various cell lines to identify those would be suitable for following functional assays, based on their sensitivity to venetoclax and expression levels of BCL-2/BCL-X$_L$. Finally, we selected several types of leukemia (Jurkat: TIB-152, EL4: TIB-39, Daudi: ccl-213, RS4:11: crl-1873) and solid tumor (A549: ccl-185, MGC-803: C6582) cells, which exhibit a relatively decent sensitivity to venetoclax and varied levels of BCL-2/BCL-X$_L$ expression (Fig. 5a, b, Supplementary Figs. 10a–d and 11a–d).

Then we started by directly treating BCL-2-overexpressing Jurkat cells with CPs. However, in contrast to venetoclax, no clear inhibitory effect was observed for these CPs (Fig. 5a). We speculated that CPs may possess a low membrane-penetrating efficiency, which limits their activity and function in cells. Trying to increase the penetration efficiency, we designed cp6 by point-mutating cp1-E10 to K10 (Supplementary Fig. 1), as more positive charges might help it engage the negatively charged cell membrane. Subsequently we synthesized the fluorescein isothiocyanate (FITC) labeled CPs, cp1-FITC and cp6-FITC (Supplementary Table 1), to trace their distribution in cells. The flow cytometry results showed that both FITC-labeled cp1 and cp6 have a relatively low penetration efficiency (Supplementary Fig. 10g, h). Therefore, we resynthesized CPs by adding on a TAT sequence (-GSGSRKKRRQRRR-), which is a widely used cell-penetrating peptide (Supplementary Fig. 10i–k). The SPR data showed that adding of the TAT sequence would not compromise the binding affinity of CPs (Table 1, Supplementary Table 5). The IC50 values of cp1-TAT in Jurkat, Daudi and El4 cells were measured as 27.5 ± 2.9 μM, 13.2 ± 0.4 μM and 28.1 ± 2.1 μM, which for each cell line were relatively higher than those of cp3-TAT (20.4 ± 2.4 μM, 7.9 ± 0.4 μM and 13.4 ± 1.6 μM, respectively). These results were roughly positively correlated with their in vitro binding affinities measured by SPR (Table 1, Fig. 5b, Supplementary Table 5). However, the IC50 values of cp2-TAT in Jurkat, Daudi and El4 were only 38.9 ± 2.7 μM, 47.9 ± 3.4 μM and 27.7 ± 4.6 μM, respectively. These values were much higher than those of cp3-TAT, especially for the Daudi cells (Fig. 5b, Supplementary Fig. 11e, f), which

might be caused by the lower membrane-penetrating efficiency of cp2-TAT.

Notably, the IC50 value of cp3-TAT in Daudi cells (7.9 ± 0.4 μM) was comparable to those of the approved drug venetoclax (9.1 ± 0.4 μM), as well as the clinical-trial drugs ABT-263 (13.5 ± 1.5 μM) and S55746 (10.0 ± 0.4 μM) (Fig. 5b, Supplementary Fig. 11f). To further compare CPs with small-molecule drugs on their inhibitory activities in cells highly expressing both BCL-2 and BCL-X$_L$ proteins, we tested them using the venetoclax-sensitive cell line RS4:11 (Fig. 5b). As expected, the IC50 values of venetoclax, ABT-263 and S55746 were 0.5 ± 0.1 nM, 0.4 ± 0.1 nM and 4.5 ± 0.1 nM, respectively. However, these CPs only exhibited very low inhibitory activities, with IC50 values of 47.8 ± 6.4 μM, 67.2 ± 3.5 μM and 16.7 ± 1.7 μM for cp1-TAT, cp2-TAT and cp3-TAT, respectively (Fig. 5b). These results, especially the over 100-fold differences between IC50 values and BCL-2-binding $K_D$ values for the CPs *versus* the similar values for S55746, imply that those TAT-tagged CPs may still have a low level of cell membrane-penetrating efficiency. Therefore, greatly improving the delivery of CPs into tumor cells would be one of the key tasks in our future study.

Next, we wanted to know whether these CPs can to some extent bypass the venetoclax-resistant mutations at a cellular level. We started by trying to construct RS4:11 cells that stably overexpress BCL-2 WT or G101V proteins, but failed despite multiple attempts. Thus instead, we selected the BCL-2 low-expressing A549 as a tool cell line to construct stable cells overexpressing BCL-2 WT or G101V proteins at similar levels as demonstrated by western blot and qPCR assays (Supplementary Fig. 10e, f). Then, we tested the inhibitory effects of venetoclax, S55746 and cp3-TAT using these A549 cells (i.e. the original cells, BCL-2 WT overexpressing cells, and BCL-2 G101V overexpressing cells). The IC50 values of venetoclax (1.1 ± 0.2 μM, 2.5 ± 0.2 μM, 4.1 ± 0.3 μM), S55746 (74.1 ± 4.5 μM, 120.2 ± 9.1 μM, 208.9 ± 9.3 μM) and cp3-TAT (45.1 ± 4.3 μM, 66.1 ± 4.6 μM, 63.8 ± 3.5 μM) were determined, respectively for the above three types of cells (Fig. 5c–e, Supplementary Fig. 10i–n). These data suggest that cp3-TAT exhibits a relatively stronger inhibitory activity on A549 cells than S55746, although still much weaker than venetoclax. Notably the overexpression of BCL-2 proteins could reduce the activities of all three inhibitors. However, further substantial activity reductions by the G101V mutation as compared with WT were observed only for venetoclax and S55746, but not for cp3-TAT. These above results imply that CPs, with suitable modifications improving their penetration efficiency, binding affinity and inhibitory activity, may provide an alternative approach to address the drug-resistant mutations of BCL-2. Meanwhile, the clinical relevance of these CPs remains to be established.

Although the poor membrane-penetrating ability of CPs severely hindered our exploration of their functional effects within tumor cells even when tagged by TAT, we still wanted to test whether the relatively weak inhibitory activities of CPs are related to the mitochondrial apoptosis pathway. To do so, we examined the release of cytochrome C in Jurkat cells (Fig. 5f, g). First, we explored the effect of BAX-inhibiting peptide (BIP)-V5, which has been known as a BAX-mediated apoptosis inhibitor through inhibiting BAX expression in certain cell lines[54,55]. We found that in Jurkat cells BIP-V5 treatment could also reduce the protein expression level of BAX (Fig. 5f). Then we compared the cytochrome C level in mitochondria upon the treatment by venetoclax or CPs, and found that it was decreased clearly by venetoclax while to a lesser extent by cp4-TAT or cp5-TAT (Fig. 5g). Interestingly, additional BIP-V5 treatment could reduce the release of cytochrome C triggered by venetoclax, cp4-TAT or cp5-TAT to a similar level (Fig. 5g). These above results suggest that the inhibitory effects of CPs on the growth of Jurkat cells might be related to the BCL-2 family-dependent mitochondrial apoptosis pathway.

To further investigate the potential pro-apoptotic effects of CPs-TAT on Jurkat cells, we employed another classic detecting

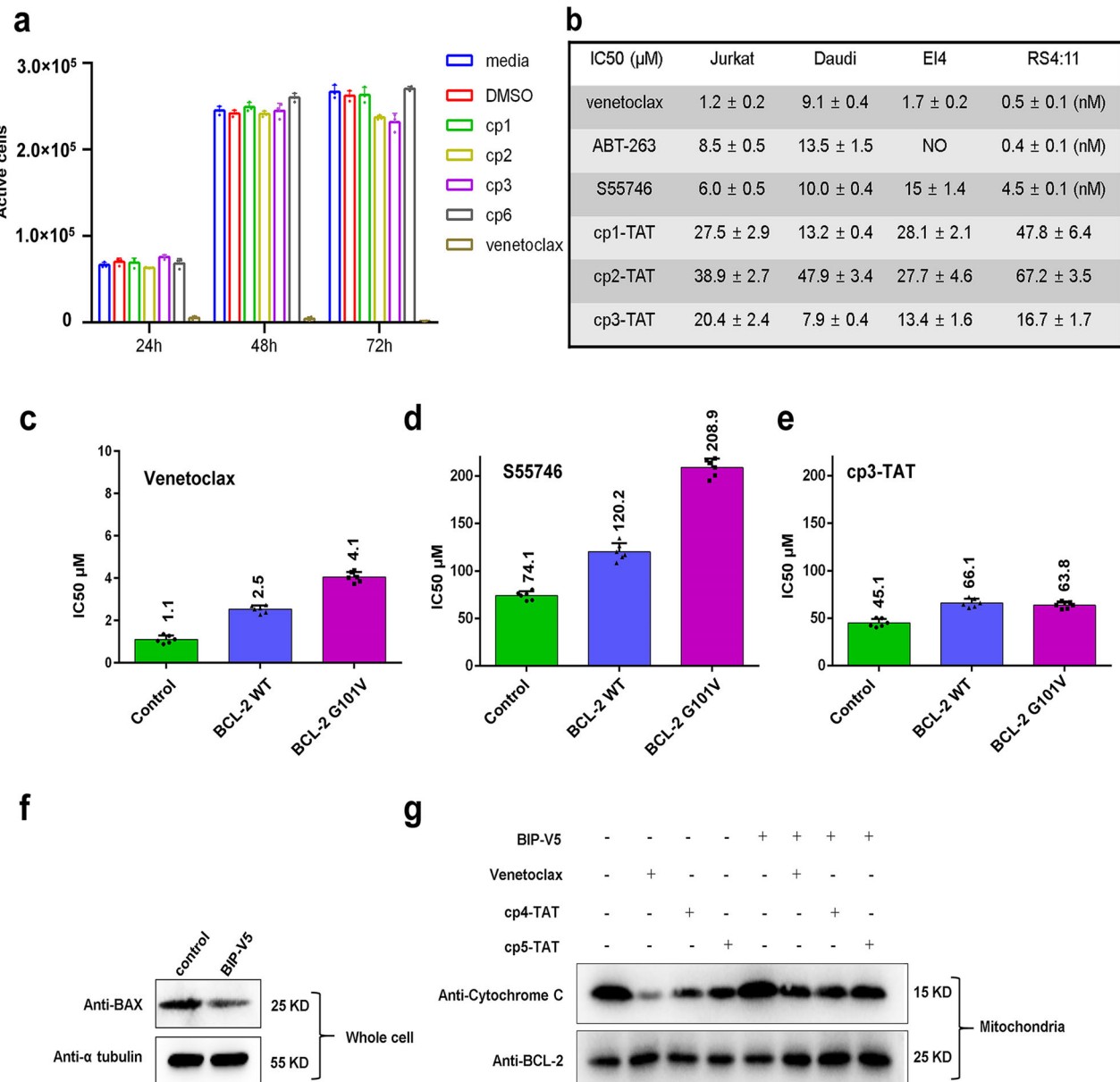

**Fig. 5 | The functional effects of CPs on different tumor cells and against BCL-2 G101V mutation. a** Effects of different CPs on the viability of Jurkat cells with venetoclax as a positive control. **b** IC50 values of venetoclax, ABT-263, S55746 and TAT-conjugated CPs each against Jurkat, Daudi, El4 and RS4:11 cells, as measured after 48 h treatment by the CellTiter-Glo assay and flow cytometry. "NO" means no inhibition detected. IC50 values of venetoclax (**c**), S55746 (**d**) and cp3-TAT (**e**) against normal A549 cells (control) and those stably overexpressing BCL-2 WT or

G101V mutant. Data are presented as mean ± S.D.; *n* = 6 biological replicates. **f** BAX inhibition assay of Jurkat cells treated by BIP-V5 at 50 μM for 24 h. The experiment was repeated 3 times independently with similar results. **g** Cytochrome C release assay of Jurkat cells treated with venetoclax (2.5 μM), cp4-TAT (20 μM), cp5-TAT (15 μM) for 12 h in the absence or presence of BIP-V5. The experiment was repeated 3 times independently with similar results.

method for apoptosis, the annexin V-FITC/propidium iodide (PI) staining and flow cytometry analysis. The results showed that after 24 h treatment of venetoclax at 2.5 μM, cp1-TAT at 30 μM, cp2-TAT at 30 μM and cp3-TAT at 20 μM, the proportions of apoptotic cells were 94.5%, 66.77%, 18.83% and 99.03%, respectively (Fig. 6a, b, Supplementary Fig. 12a). Obviously, CPs especially the cp3-TAT could induce apoptosis of Jurkat cells. To assess whether the pro-apoptotic effects of CPs involved the mitochondrial pathway, we tested the mitochondrial membrane potential (ΔΨm). In normal cells, JC-1 dye aggregates in the mitochondrial matrix and produces strong red fluorescence (Supplementary Fig. 13a). Relative ΔΨm of Jurkat cells slightly decreased after treatment of cp1-TAT and cp2-TAT, as indicated by the weakened red fluorescence (Supplementary

Fig. 13c, d). Meanwhile the red fluorescence almost disappeared and green fluorescence enhanced, after cells were treated with veneto-clax or cp3-TAT (Supplementary Fig. 13b, e). We also quantitatively analyzed the changes of ΔΨm by flow cytometry. Data showed that the ΔΨm after 4 h treatment of 1‰ DMSO, TAT control, venetoclax, cp1-TAT, cp2-TAT and cp3-TAT were 20.84, 27.37, 0.1, 4.15, 11.43 and 1.53, respectively (Fig. 6c, d, Supplementary Fig. 12b). The influence of ΔΨm by different inhibitors were consistent with immuno-fluorescence staining and apoptosis results (Fig. 6a, b, Supplementary Fig. 13), with cp3-TAT showing the strongest activity among three CPs (although still much weaker than venetoclax). These results imply that CPs may inhibit the growth of tumor cells through the mitochondrial apoptosis pathway. However, given the large

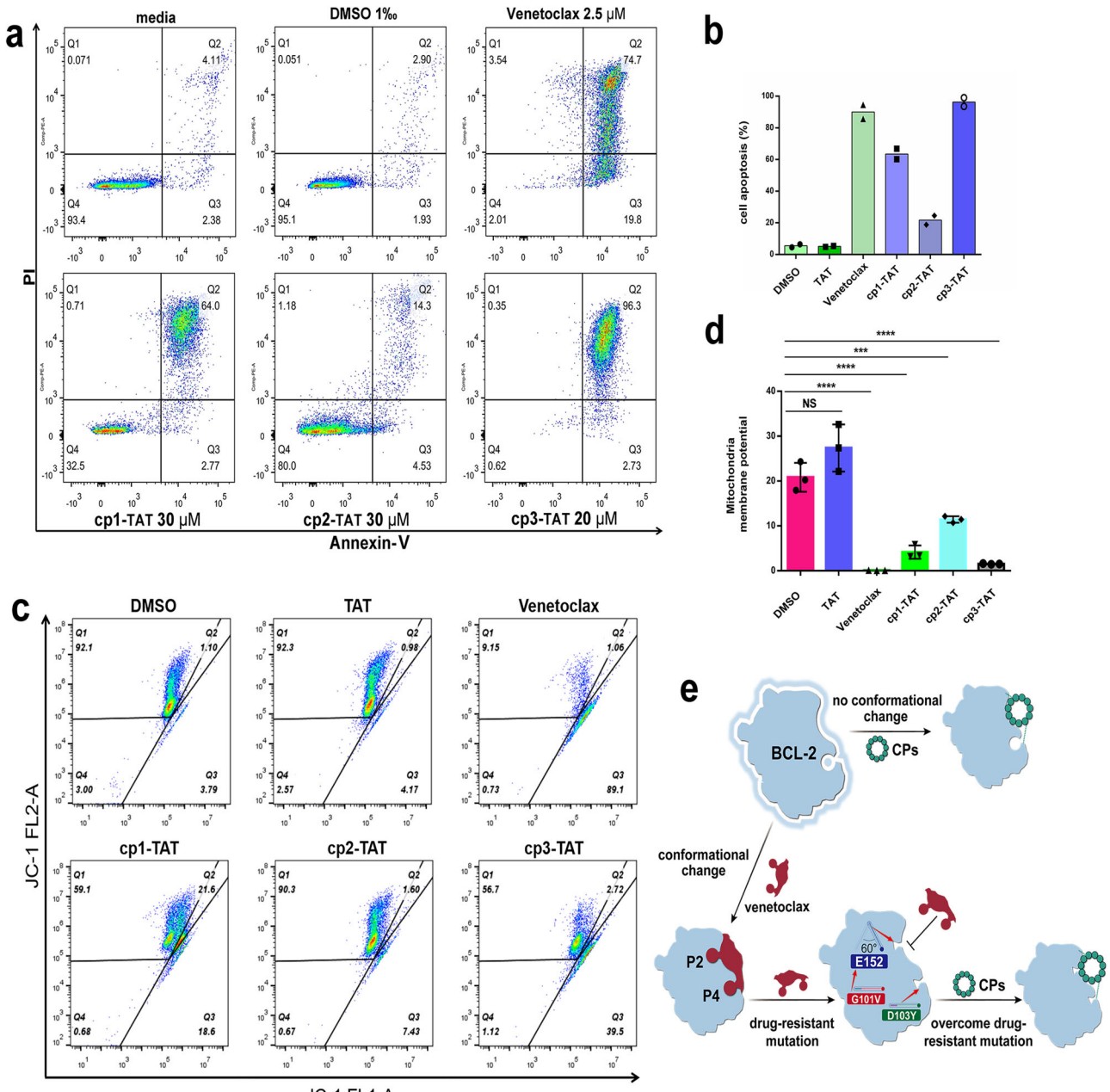

**Fig. 6 | The pro-apoptotic mechanism of CPs in killing tumor cells.**
**a** Representative results from the apoptosis assay of Jurkat cells using Annexin V-FITC/PI staining. Q4 region contains Annexin V⁻/PI⁻ cells (live cells); Q3 shows Annexin V⁺/PI⁻ cells (early apoptotic or apoptotic cells); Q2 represents Annexin V⁺/PI⁺ cells (late apoptotic cells); Q1 contains Annexin V⁻/PI⁺ cells (mechanical damaged cells). **b** Quantitative analysis of Jurkat cell apoptosis after 24 h treatment with different inhibitors. **c** The mitochondrial membrane potential (mtΔΨ) analysis of Jurkat cells after 4 h treatment with different inhibitors. **d** Statistical analysis of mtΔΨ (Q1/Q3). Data are presented as mean ± s.d.; $n = 3$ biological replicates; one-way ANOVA with Dunnett's multiple comparisons test. The calculated $P$ values were $P = 0.053$, $P < 0.0001$, $P < 0.0001$, $P < 0.001$ and $P < 0.0001$ for TAT, venetoclax, cp1-TAT, cp2-TAT and cp3-TAT, respectively. NS no significant difference. **e** A diagram showing the special binding mode of CPs and how CPs might overcome drug-resistant mutations.

differences between CPs' direct binding affinities to BCL-2/BCL-X_L proteins (Table 1) and their cellular activities (Figs. 5 and 6), we could not fully rule out the possibility that certain off-target effects are also involved in the inhibition by CPs on these tumor cells.

## Discussion

Targeting the PPIs between BCL-2 family proteins for cancer therapy has long been an attractive strategy. It is known that traditional small-molecule inhibitors targeting the BH3-only binding interface of BCL-2/BCL-X_L with high affinities often induce conformational changes of the hydrophobic surface groove[56]. Moreover, we analyzed all the complex structures of BCL-2 and BCL-X_L bound with traditional inhibitors in the PDB database, and found that these inhibitors all induce conformational changes of α3 and α4 helices to fit into the P2 or P4 pockets of two proteins (Supplementary Fig. 5). This common mechanism of action among current small-molecule drugs may lead to their inevitable compromise to the same set of clinical mutations. Therefore, it is urgent to develop a different type of inhibitors to potentially combat these mutations.

As previously speculated from the complex structure of BCL-2 G101V-venetoclax[31], ligands with scaffolds that don't need insert deeply into P2 or P4 pockets may tightly bind both to BCL-2 WT and to the

G101V mutant. In addition, our structural analysis and MD simulation results imply that driving conformational changes of a network of key residues involved in the flexible pockets P2 (F104, F112, M115, E152), P3 (G101, F104) or P4 (D103) may be the common approach behind clinical drug resistance. Unlike known inhibitors, the CPs we discovered can bind directly onto the flat surface of the BCL-2 hydrophobic groove, without inducing conformational changes dramatically. Consistent with their unique binding mode, these CPs demonstrate similar binding affinities and inhibitory activities towards both the BCL-2 WT and G101V mutant. Therefore, these CPs may have a potential to overcome the challenges faced by traditional inhibitors targeting BCL-2 family proteins, and may even provide clues for restraining PPIs of other related targets.

Until now, the challenge of obtaining inhibitors targeting BCL-2 or BCL-$X_L$ with a very high selectivity has not been fully resolved. Although some selective inhibitors like venetoclax and S55746 targeting BCL-2, and WEHI-539 targeting BCL-$X_L$ were discovered[13,31,56], the structural mechanism of their selectivity had not been very clear. Here, we identified the single-residue discrepancy (BCL-2 D111/ BCL-$X_L$ A104) as a "switch" that may regulate the binding specificity of CPs in two different ways. First, by forming a hydrogen bond with D111, CPs (like cp1) can gain a high selectivity to BCL-2 over BCL-$X_L$. Second, due to the side-chain difference between A104 and D111, BCL-$X_L$ and BCL-2 exhibit a different binding interface at this position. Using this different interface, we obtained cp3 and cp5, which preferentially and tightly bind to BCL-$X_L$ by fitting the surface around A104 with their special P6 amino-acid residue. On the other hand, the BCL-2 D111 residue may be considered as a special loading dock to design covalent or other type of inhibitors specifically targeting BCL-2, in a similar way to the recently reported inhibitors of KRAS (G12D)[57,58].

To further examine the potential of this intriguing "switch", we analyzed the crystal structure of BCL-2 in complex with S55746[59]. S55746 inserts its 4-hydroxyphenyl moiety into the P2 pocket of BCL-2 much deeper than CPs, and it also forms a hydrogen bond with D111 (Supplementary Fig. 14). Consistent with previously reported data[13], we measured the binding affinities of S55746 to BCL-2 and BCL-$X_L$ as $0.006 \pm 0.002 \, \mu M$ and $0.175 \pm 0.042 \, \mu M$ (Table 1), implying a nearly 30-fold selectivity to BCL-2. However, the binding affinity of S55746 to BCL-2 mutant D111A is $0.006 \pm 0.003 \, \mu M$, similar to BCL-2 WT (Table 1). These results suggested that the effect of this special BCL-2/ BCL-$X_L$ "switch" would only be remarkable for inhibitors that don't employ the deep pocket P2 for association, like the cp1. In addition, we also tested the selectivity of CPs against other anti-apoptotic BCL-2 family proteins, and found that they could bind to BCL-w, but not to MCL-1 and BFL-1 (Table 1). Then, we analyzed key residues in the hydrophobic surface groove of all anti-apoptotic members of BCL-2 family, and found that these residues in BCL-w show a high similarity to BCL-2 and BCL-$X_L$, but those in MCL-1 and BFL-1 do not (Supplementary Table 6).

The most critical factor that hinders the application of peptide drugs might be their limitation of cell-membrane permeability, the same difficulty we encountered in this work for CPs. The molecular weights of our CPs exceed 1000 Da, much beyond the "Lipinski's rule of five" for small-molecule drugs[60]. Researchers have found that those naturally permeable macrocycles (e.g. cyclosporine A and griselimycin) possess some common features, such as low polar surface area, lack of unsatisfied hydrogen bond donors, and conformational switch between different structures in aqueous and membrane environments[61–63]. The structural characteristics and distinctive binding mechanism of our CPs may lay the foundation for enhancing their membrane-penetrating efficiency, target selectivity, and inhibitory activity. Our study has the potential to contribute to the future development of selective inhibitors that target BCL-2 and/or BCL-$X_L$.

## Methods

### Construction of recombinant plasmids

The DNA fragments encoding human BCL-$X_L$ (residues 1–210, with 27–81 and 210–233 removed), BCL-w (residues 6–152), MCL-1 (residues 152–308) and BFL-1 (residues 1–151) as previously reported[18] were cloned into a pET-22b vector (Invitrogen), respectively. A chimeric human BCL-2 (including residues 1–50 and 92–207) was constructed, by removing the internal long loop (residues 51–91) and replacing its residues 35–50 with residues 33–48 of BCL-$X_L$ to solubilize the protein[14], and then it was cloned into a pET28a-Sumo vector (Invitrogen). All BCL-2 mutants (G101V, F104L, Y103D, D111A) were generated by mutagenic PCR, according to the QuikChange (Stratagene) protocol.

### Protein expression, purification and concentration analysis

The proteins were expressed in the *E. coli* BL21 (DE3) strain (Novagen), which was initially cultured at 37 °C until OD600 reached about 0.6 to 1. Next, 0.4 mM isopropyl β-D-thiogalactoside (IPTG, Sigma) was added to induce protein expression at 16 °C overnight. For purification, the supernatant from cell lysate was first loaded onto a Ni-NTA column (Cytiva), followed by further separation using three columns HiLoad 16/600 Superdex 75 pg (Cytiva), RESOURCE Q 6 mL (Cytiva), and Superdex 200 Increase 10/300 GL (Cytiva), respectively. Finally, the purity of the proteins reached above 95%. The proteins were concentrated to 6–10 mg/mL for crystallization and other assays. All samples were flash-frozen in liquid nitrogen and stored at −80 °C.

### Surface plasmon resonance (SPR) assays

SPR assays were performed using a Biacore T200 instrument in running buffer consisting 10 mM PBS, 50 μM EDTA and 0.05% Tween-20 at 25 °C. Biotinylated proteins were immobilized on a CAP sensor chip (Cytiva, 28920234). Serially diluted concentrations of peptides were tested with multiple cycles (25-12800 nM, depending on the peptides) at a flow rate of 30 μL/min. The binding affinities were quantified by the Biacore evaluation software using 1:1 binding model. The reproducibility of the assays was confirmed by performing replicate determinations.

### Crystallization, data collection, processing and structure refinement

Crystals of protein-CP complexes including BCL-$X_L$-cp1, BCL-$X_L$-cp2, BCL-$X_L$-cp3, BCL-2-cp1, BCL-2 G101V-cp1 and BCL-2-12M-cp2 were obtained at 16 °C by hanging-drop vapor diffusion. The droplets consisted of a 1:1 (v/v) mixture of proteins and the well solution containing one of the following recipes: 0.1 M HEPES sodium pH 7.5, 2.0 M Ammonium formate; 0.2 M Ammonium acetate, 0.1 M Bis-Tris pH 6.5, 45% MPD; 30% (v/v) PEG 400 100 mM Tris base/ Hydrochloric acid pH 8.5, 200 mM Magnesium chloride; 0.1 M Bis-Tris pH 6.5, 2.0 M Ammonium sulfate; 0.2 M Calcium acetate, 20% PEG3350; 0.5 M Ammonium sulfate, 0.1 M Sodium citrate tribasic dihydrate pH 5.6, 1.0 M Lithium sulfate monohydrate. Crystals appeared after 36-48 h and were ready for data collection in about 14 d. The crystals were flash-frozen in liquid nitrogen to −170 °C with 25% glycerol as cryoprotective liquid. The diffraction data were collected at 100 K under the synchrotron radiation at beamlines BL19U1 and BL18U1 of the Shanghai Synchrotron Radiation Facility (SSRF). The data sets were integrated and scaled with the HKL3000 package[64]. The structures were determined by molecular replacement with the structure of BCL-2-venetoclax and BCL-$X_L$-WEHI-539 (PDB IDs: 6O0K and 3ZLR) as the initial search models with the program Phaser[65]. The programs Refmac5 and Coot in the CCP4 suite[66] were used for the refinement and model building. The Ramachandran statistics, generated by Coot, are 95.52%/0, 98.53%/0, 94.33%/0, 97.76%/0, 98.61%/0, and 97.54%/0 for 7Y90 (BCL-2-cp1), 7Y8D (BCL-$X_L$-cp1), 7Y99 (BCL-$X_L$-cp2), 7YA5 (BCL-2-G101V-cp1), 7YAA (BCL-$X_L$-cp3) and 7YB7 (BCL-2-12M-cp2),

respectively. The statistics for data processing and structure refinement are shown in Supplementary Table 2.

## Molecule dynamics (MD) simulation

In order to explain the effects of venetoclax-resistant mutations on ligand binding from structural and dynamic perspectives, we used the MD simulation to compare the BCL-2 WT and G101V mutant with or without ligand binding. Crystal structures of these two proteins and the protein-cp1 complex resolved in this work were adopted as the initial structures for the ligand-free and BCL-2-cp1 simulations, respectively. The initial structures for the BCL-2-venetoclax simulations were derived from PBD IDs, 6OOK and 6OOL for WT and G101V mutant, respectively. In all cases, the protonated nitrogen of the imidazole ring of histidine residue was at the Epsilon position, and missing side-chains were added using the PDBFixer tool (https://github.com/openmm/pdbfixer). Water molecules or other buffer solvents in the crystal structure were removed.

The charmm36m force field[67] was chosen as the interaction parameter for the protein. The TIP3P water[68] (and a cubic box with a 10 Å boundary) was selected as the solvation model. Buffer of 0.05 M NaCl was added to ensure the system is electrically neutral. We used the CGenFF[69] for the bonded and van der Waals parameters of ligands venetoclax and cp1. The partial atomic charges were calculated by the RESP method[70] with ORCA[71] and Multiwfn[72]. The initial position of the ligand extracted from the complex mentioned above was used for geometry optimization at the B97-3c level. Then charges were fit at the B3LYP/G level with D3 dispersion correction[73] and SMD solvation model[74] (parameters shown in Supplementary Data 1).

All simulations were performed using OpenMM (version 7.7)[75]. Simulations were conducted at 300 K and 1 atm using a Monte Carlo barostat. To get enough sampling that captures side-chain allosteric, a tailored Hamiltonian replica exchange (HREX) protocol was used to accelerate side-chain conformational changes associated with G101V mutation and ligand binding (script shown in Supplementary Data 1). Specifically, all the torsion potential terms on the side chains near the ligand (D103, F104, R106, D111, F112, M115, V133, E136, L137, R139, N143, R146, E152, F153, Y202) in the complex were scaled according to the following Eq. (1):

$$U'_{torsion} = scale_n \times U_{torsion} = \frac{T_0}{T_n} \times U_{torsion} \qquad (1)$$

where $n$ be the replica index, $U_{torsion}$ be the selected side-chains torsion potential term.

Each system had 8 replicas with different scale factors from 1.0 to 0.7, equivalent to the temperature from 300 to 430 K (see Supplementary Table 7 for details). The simulation time for each replica was 200 ns. The exchange was attempted every 4 ps. And the final exchange probability was between 0.2 and 0.3.

The first replica (scale factor = 1.0) was chosen for the chi1 or chi2 angle distribution analysis. The chi1 angle calculation was done by the mdtraj package[76]. Correlation analysis of the chi angle used all the trajectories mentioned above to capture as many conformational transition events as possible. Figures related to chi1 or chi2 were plotted by the matplotlib package (https://matplotlib.org/).

The solvent-accessible surface area (SASA) calculation was conducted with MDTraj[76]. The full documentation and examples are available on the project home page (http://mdtraj.org), and its development is hosted by GitHub (http://github.com/mdtraj/mdtraj).

## Structure-based phage screening for improved CPs

To obtain a phage library that displays peptides: CPXRYXWDXXC (X represents any of the 20 natural amino acids), oligonucleotides encoding peptides (5'-TCGCGGCCCAGCCGGCCATGGCATGTCCGNN KCGCTATNNKTGGGATNNKNNKTGTGCGGCCGCAAGGTGCGCCGGT

G-3') were co-incubated with the reverse primer (5'-CACCGGCG-CACCTTGCGGCCGC-3') in extended system containing Klenow fragment. The double-stranded products of DNA and phagemid vector pCANTAB 5E were digested by Sfi I (50 °C, 8 h) and Not I (37 °C, 8 h). Then, the purified Sfi I/Not I-digested DNA fragments and phagemid vector were ligated by T4 DNA ligase. The ligation mixture was transformed into *E. coli* TG1 competent cells. The cells were plated on 2× YT/ampicillin agar plates and incubated at 37 °C overnight. The phage library capacity was 1.6 ×10⁷, which was evaluated by measuring the total number of colonies. Finally, the colonies were scraped from the plates and used for further phage production and purification. The methods of cyclization of peptide library, biotinylation of protein, phage screening, synthesis and purification of peptides, and ADT cyclization of peptides were the same as described previously[44,77].

## Cell culture

The cell lines Jurkat (TIB-152), El4 (TIB-39), Daudi (ccl-213), RS4:11 (crl-1873), A549 (ccl-185), HepG2 (hb-8065) and PANC-1 (crl-1469) were purchased from ATCC (American Type Culture Collection), and MGC-803 (C6582) was purchased from Beyotime (https://www.beyotime.com/product/C6582.htm). Jurkat and EL4 cells were cultured in RPMI-1640 medium (BasalMedia, L210KJ), supplemented with 10% inactive FBS (ExCell), 1 mM Sodium pyruvate (Gibco, 11360070), 10 mM Hepes pH 7 (Gibco, 15630080), 50 μM β-mercaptoethanol (Sigma), and 1% penicillin-streptomycin (Gibco, 2321118). Daudi and RS4:11 cells were cultured in RPMI-1640 medium (HyClone, SH30809.01) and 10% FBS (Gibco, 10099-141), whereas HepG2, MGC-803 and PANC-1 cells were cultured in MDEM (HyClone, SH30243.01). Cells were maintained at 37 °C under a humidified, 5% (v/v) CO₂-containing atmosphere.

## Cell viability assay

Cell viability was measured using CellTiter-Glo® 2.0 Reagent (Promega, G9242). Different cells (Jurkat, El4, Daudi, RS4:11 and A549) were treated with CPs, venetoclax, S55746, or ABT-263 in different concentrations (0–100 μM, serial dilution and 6 replicates per gradient). After different inhibitors were added into serum-free medium and cultured for 4 h (RS4:11 for 6 h), FBS were supplemented to 10% and then incubated totally for 24 h or 48 h. Finally, the IC50 values were calculated based on measurements using a microplate reader (Thermo, Varioskan LUX). Statistical analysis was performed using software GraphPad Prism 7.0.

## Measurement of mitochondrial membrane potential

For the mitochondrial membrane potential (ΔΨm) analysis, the mitochondrial membrane potential assay kit with JC-1 dye (Beyotime, C2006) was used according to the manufacturer's protocol. In cells with high ΔΨm, JC-1 aggregates and shows red fluorescence; while in cells with low ΔΨm, the JC-1 monomer shows green fluorescence. The value of ΔΨm was expressed as the ratio of red fluorescence intensity over green fluorescence intensity. After treated with JC-1, fluorescence intensity of Jurkat cells was tested by flow cytometry (BD FACSAria Fusion) and fluorescence microscopy (NIKON Ti-E), respectively. All experiments were performed in triplicate. Statistical analysis was performed using software GraphPad Prism 7.0.

## Flow cytometry analysis

Jurkat cells were seeded to 6-well plates at a density of 4 ×10⁵ cells per well. Cells were incubated in the presence of DMSO, venetoclax at 2.5 μM, cp1-TAT at 30 μM, cp2-TAT at 30 μM, or cp3-TAT at 20 μM in serum-free RPMI medium for 4 h. After being incubated totally for 24 h, cells were washed once with PBS and stained with 100 μL binding buffer containing 5 μL Annexin V-FITC and 10 μL PI for 15 min, followed

by addition of 400 µL binding buffer to detect cell apoptosis with FCM (BD FACSAria Fusion). Statistical analysis was performed using software FlowJo V10 and GraphPad Prism 7.0.

## Cytochrome C release
Jurkat cells at $2 \times 10^6$ cells/mL were incubated in the presence of DMSO or 50 µM BIP-V5 inhibitor[54] (Yuanye Biotechnology Co., Ltd, 579492-81-2) for 24 h. Two groups of above treated cells were each divided into four groups and incubated in the presence of DMSO, venetoclax at 2.5 µM, cp4-TAT at 20 µM, or cp5-TAT at 15 µM in serum-free RPMI medium for 4 h. Then the medium was supplemented with FBS to 10% and incubated for another 8 h. After being treated by venetoclax or CPs for totally 12 h, cell were processed with the Mitochondrial Separation Kit (Beyotime, C3601) to extract mitochondria. Cytochrome C in mitochondria and BAX proteins in the whole cells were detected by Western blot.

## Fluorescence polarization (FP) assays
Both fluorescence polarization binding assay and competition assay were performed in 1× PBS (pH 7.4), and fluorescence anisotropy was measured in a black opaque 96-well microplate (PerkinElmer) using an Infinite® 200 PRO multimode microplate reader ($E_{ex}$=485 nm, $E_{em}$ = 535 nm, TECAN). The binding and competition polarization data were fitted using Origin 2021.

Based on the sequence of cp2/cp3, peptides were synthesized and labeled fluorescently by conjugating the amino group in its C-terminal lysine with fluorescein isothiocyanaye (FITC), and they were modified by ClAc-3 (cp2-F: CPIRYEWDEFCG{K(FITC)}; cp3-F: CPIRYPWDVECG{K(FITC)}). The peptide solution was prepared prior to use by diluting the 1.0 mM stock solution in DMSO, using 1× PBS (the final concentration of DMSO was less than 1%).

For the FP binding assay, cp2-FITC/cp3-FITC solution at the final concentration of 20 nM was added to protein solution ranging from 60 nM to 1000 nM. Samples were incubated at room temperature for 10 min before measurement. Dissociation constant ($K_D$) values were determined by nonlinear regression analysis of anisotropies versus the concentrations of protein using the Eq. (2)[78].

$$y = A_0 + (A_{max} - A_0) \times \frac{(x + c + K_D) - \sqrt{(x + c + K_D)^2 - 4xc}}{2c} \quad (2)$$

$x$: the concentration of protein, $y$: the measured fluorescence anisotropy, $c$: the concentration of fluorescent peptide, $A_0$: the bottom plateaus of anisotropy, $A_{max}$: the anisotropy signal in the presence of saturating concentrations of protein, $K_D$: the dissociation constant.

For the FP competition assay, cp2/BCL-2 complexes (20/400 nM) and cp3/BCL-$X_L$ complexes (20/600 nM) were prepared, followed by treatment with peptides ranging from 1.0 nM to 30.0 µM. Samples were incubated at room temperature for 10 min before measurement. The polarization data were fitted[79] with the Eq. (3) in Origin, based on one-site competitive model. The inhibition constant $K_i$ was calculated[80] by the Eq. (4).

$$y = A_{max} + \frac{A_{max} - A_0}{1 + 10^{(x - \log EC_{50})}} \quad (3)$$

$$K_i = EC_{50} / \left( \frac{L_{50}}{K_D} + \frac{P_0}{K_D} + 1 \right) \quad (4)$$

$x$: the log molar concentration of the oxidized peptides, $y$: the measured fluorescence anisotropy, $A_{max}$: the top plateaus of anisotropy, $A_0$: the bottom plateaus of anisotropy, $EC_{50}$: the concentration of peptide at 50% inhibition, $L_{50}$: the concentration of the FITC-labeled peptide at 50% inhibition, $P_0$: the concentration of the free protein at 0% inhibition, $K_D$: the dissociation constant of fluorescent peptide to protein.

## Plasmids, retrovirus production and cell infection
Lentivirus packaging vectors pMDLg/pRRE, pRSV-Rev, and pVSV-G were obtained from Prof. Han Jiahuai's lab (Xiamen University). Lentiviruses were produced using 3rd-generation lentivirus packaging plasmids transiently transfected into HEK293T cells with the constructs of interest using 0.25 M $CaCl_2$. Media was changed after 16 h, and supernatants containing infectious virus particles were harvested 24 h later. A second viral harvest was made following a further 24 h incubation with fresh media. Virus containing supernatant was filtered through a 0.45 µM filter, and stored at 4 °C or −80 °C until being used. Typically, A549 cells were seeded into 10 cm cell culture dishes at $4 - 8 \times 10^5$ cells/cm². An equivalent volume of virus containing culture medium was added along with polybrene (Sigma) to a final concentration of 10 µg/mL. Cells were incubated at 37 °C for 20 h, before they were washed and resuspended in fresh culture media. The expression level of BCL-2 was measured by qPCR and western blot methods.

## Western blot
Cell lysates containing equal amounts of total proteins were denatured, subjected to electrophoresis on SDS-PAGE gels, and transferred onto nitrocellulose membranes. The blots were blocked with 5% skim milk and then incubated at 4 °C overnight with the corresponding primary antibodies against BCL-2 (dilution 1:1000, Abcam, ab194583), BCL-$X_L$ (dilution 1:1000, Abcam, ab32370), anti-Flag (dilution 1:1000, Sigma, F1804), cytochrome C (dilution 1:1000, Beyotime, AC909), BAX (dilution 1:1000, Beyotime, AB026), α-Tubulin (dilution 1:3000, Abcam, ab7291), or β-Actin (dilution 1:3000, Abcam, ab8227). After incubation with the secondary antibody of Goat anti-Rabbit IgG (dilution 1:2000, Abcam, ab6721) or Goat anti-Mouse IgG (dilution 1:2000, Abcam, ab6789) for 1.5 h at room temperature, bands on the membranes were visualized using an ECL kit (Pierce, 32106).

## qPCR
Total RNA was prepared by using the TriPure Isolation Reagent (Roche, 11667157001). Reverse transcription was conducted with RevertAid First Strand cDNA Synthesis Kit (Applied Biological Materials Inc., G592). The sequences of primers used for real-time PCR assays were F: 5′-GCCACCTGTGGTCCACCT-3′, R: 5′-CTGAAGAGCTCCTCCACCAC-3′ for BCL-2 and F: 5′-CAAGGTCATCCATGACAACTTTG-3′, R: 5′-GTCCACCACCCTGTTGCTGTAG-3′ for GAPDH. Real-time PCR was performed in triplicate with SYBR Green qPCR Master (Roche, 25325100). The expression of genes was normalized to the GAPDH level.

## Reporting summary
Further information on research design is available in the Nature Portfolio Reporting Summary linked to this article.

# Data availability
The refined structural models and corresponding structure-factor amplitudes have been deposited in the PDB database under accession codes 7Y90 (BCL-2-cp1), 7Y8D (BCL-$X_L$-cp1), 7Y99 (BCL-$X_L$-cp2), 7YA5 (BCL-2-G101V-cp1), 7YAA (BCL-$X_L$-cp3) and 7YB7 (BCL-2-12M-cp2). All the structures cited in this work are also available under accession codes 1MAZ, 2XA0, 6OOK, 1GJH, 3ZLR, 4QVE, 2YXJ, and 1AF3. Source data are provided with this paper.

# Code availability
The MD simulation parameter files are available in Supplementary Data 1.

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

## Acknowledgements

We thank Xiaoju Li, Haiyan Yu, Xiaomin Zhao, Yuyu Guo and Changbin Liu from Core Facilities for Life and Environmental Sciences of Shandong University for their assistance in X-ray diffraction and flow cytometer analysis. We also thank the staff of BL19U1, BL18U1 and BL02U1 beamlines of SSRF (Shanghai Synchrotron Radiation Facility) for their assistance during data collection. This work was supported by grants from Shandong Provincial Natural Science Foundation, China (ZR2021JQ30), University Innovation Group Program of Jinan (202228080), National Natural Science Foundation of China (22177063), and the Taishan Scholars Project of Shandong (tsqn201909004) to D.W., grants from National Natural Science Foundation of China (31800664), China Postdoctoral Science Foundation (2021M691949), and Shandong Postdoctoral Innovation Project (202102013) to F.L., as well as grants from National Natural Science Foundation of China (22174119) and the Fundamental Research Funds for the Central Universities (20720210001 and 20720220005) to C.W.

## Author contributions

F.L., C.W. and D.W. conceived the study; C.W. and D.W. supervised the experiments; F.L., C.L., Y.H., X.C., X.S. and X.D. constructed plasmids; F.L. and C.L. purified proteins, carried out crystallization, collected X-ray diffraction data, solved and refined the structures; F.L., J.L. and Z.L. designed, synthesized and purified CPs; F.L., C.L., W.C., D.X. and J.Z. conducted cell culture and drug sensitivity tests; F.L., X.P. and Sheng W. analyzed MD simulation data; F.L. and Sen W. analyzed mitochondrial membrane potential data; F.L., C.W. and D.W. wrote the manuscript with inputs from all authors.

## Competing interests

The authors declare no competing interests.
