## [Peer Review File · Nature Communications]

Cyclic peptides discriminate BCL-2 and its clinical mutants from BCL-XL by engaging a single-residue discrepancyREVIEWER COMMENTS

Reviewer #1 (Remarks to the Author):

General comments

This manuscript by Li et al describes the development of cyclic peptides that target BCL-2 and BCL-XL, two members of the pro-survival protein family. By extensively analysing their binding mode via structural biology and molecular dynamics, they design peptides with specificity for one or the other. The paper provides some rationale design towards peptides which retain affinity for mutant versions of BCL-2 that have been identified in the clinic after venetoclax treatment.

Venetoclax has completely changed how certain cancers are treated, and continues to be tested against an ever-increasing range of tumours, as single agent or in combination. It remains the only BH3 mimetic drug currently approved. Others, targeting either BCL-2, BCL-XL or MCL1 are being clinically tested. As with all anti-cancer drugs used in the real world, clinicians have started cataloguing mutations conferring resistance to venetoclax. This is a critical issue as patients relapsing due to these altered forms of BCL-2 have little therapeutic options available. Solutions to this problem, with small molecules equally targeting all forms of BCL-2 or specifically the mutant forms, would have great therapeutic impact as second line treatment.

The data presented by Li et al aims at providing a solution to this issue and would therefore be an important advance in the field. Unfortunately, the paper falls short in three main areas and wouldn't be publishable in its current form in Nature Communications:

- Firstly, while the observations presented by the authors are interesting in the context of the cyclic peptide designed in this study, they are highly context specific and most likely not applicable to other small molecules. This is a conclusion that the authors themselves arrive at in their conclusion/discussion (P19, line 474-475). The broad applicability of their results is therefore arguable.
- The biological activities presented (both binding and cellular) are far from therapeutically relevant. The SPR data looks robust (although more about this in my third point), but is still at least one log away from what is observed with endogenous BH3 only proteins and validated BH3 mimetics (single digit nanomolar or even picomolar). Moreover, the differences in selectivity observed are in some cases small and in ranges of activity that would be difficult to translate into a therapeutic effect in vivo. In particular, the on-target mechanism of the peptides with activity with IC50 values above 40 μ M is questionable.
- The data in some places is not convincing. The SPR experiments are well conducted but no replicates (while mentioned in the text) are shown or included in the statistical calculation of the KD values. Similarly, statistics are missing on key figures with cellular data which diminish the impact of the conclusions drawn by the authors (see specific comments for examples). The demonstration that the cyclic peptides are acting by triggering mechanism-based apoptosis is also not satisfactory. The peptides at concentration above 40 μ M are provoking apoptotic cell death but the authors do not present any functional data proving that this apoptotic cell death is indeed due to specific interaction to BCL-2 or BCL-XL. There are plenty of cellular models available where various pro-survival proteins are deleted that

can help with this (see more details in specific remarks below).

Specific remarks

Page 4, line 72 “it is actually very difficult to design BCL-2-selective inhibitors”: this is not quite correct. Apart from ABT-737, ABT-263 and related compounds, most other advanced and validated BH3-mimetics have remarkable selectivity for their respective targets. What is true however, is that it is difficult to design selectivity "de novo".

Page 8, line 176-178, comment on the novelty of binding mode: this is somewhat of an overstatement. The conformational changes observed with most validated BH3 mimetics is probably a key element in their very high affinities, and in some cases selectivity, for their target. Also, it is not clear whether the analysis provided by the authors includes peptido-mimetics compounds or other peptides derived from BH3-only proteins.

Page 9, line 201-202 “...BCL-2 G101V mutant was about 1.1 μM (close to the 0.8 μM for BCL-2 WT),...”: While these two values are indeed close, comparison between them is difficult since SD or SEM are not included in Table 1. It is not clear whether this data was calculated from several independent experiments. In addition, in other places, the authors draw difference conclusions from relatively similar differences between values.

Page 14, lines 306 to 312: These conclusions are true but the changes they are associated with are small in comparison to the binding affinities that would be measured for peptides derived from the BH3-only proteins, so it not clear that this the observation is important.

Page 14, lines 323-324: “...selectivity of CPs are not simply defined by the size of their interfaces to proteins, yet likely determined by the very nature of CP residues,..”: this is an obvious statement that is true for any binder to any proteins.

Page 16, lines 370-372: The binding affinity of CP1-TAT and CP3-TAT should be measured to be sure these constructs still have the same binding as their parent compounds.

Page 16, lines 373 “...roughly consistent with their in vitro binding affinities...”: what do the authors mean by this statement?

Page 17, line 378 “...was even lower than those of the approved drug venetoclax...”: this data (IC50 values close) is not presented with any SD or SEM. Comparison between these values is not possible, especially to draw such strong conclusion. Overall, these are fairly similar values.

Page 17, line 381 “...they may exhibit potent inhibitory functions as small-molecule drugs...” and P19, line 449: the data presented is not in line with this statement. In particular, the authors haven't chosen cell lines sensitive to BCL-2 or BCL-XL inhibition to start with: e.g. RS4,11 for BCL-2 selective compounds where inhibitors have low nanomolar inhibitory activity. These established cell lines with known

sensitivity, combined with validated compounds should be used to assess the actual level of activity of the CPs. The data presented (~ 1 μ M) is not suggestive of “potent inhibitory functions”. In addition, the low cell membrane permeability is not sufficient to explain this low activity since the small molecules BH3 mimetic compounds are not particularly potent either.

P17, line 392-393: similar comment here. These IC50 values are well above what would be expected for on-target activity in cells dependent on BCL-2 for example.

P17, line 395: Similar comment and at that level of activity, it is possible that the killing observed is not due to selective BCL-2 inhibition.

P17, line 399 and P18, line 418-419 “...another hallmark detecting method for apoptosis,...”: while these are classical phenotypic hallmarks of apoptosis, there are not providing information demonstrating on target/mechanism based activity. For lists of criteria for mechanism-based activity, refer to the following reviews: Lessene et al, Nat Rev Drug Discover, 2008; Soderquist et al, Mol Canc Ther, 2016 and Villalobos Ortiz et al, Cell Death Diff, 2018.

P19, line 437 “...onto the flat surface of the hydrophobic groove, without inducing conformational changes of the α 3 and α 4 helices...”: considering the relatively weak level of affinity and very weak cellular activity, it is possible that the lack of conformational changes to accommodate these peptides is also a drawback. The true comparison here would be with peptides derived the BH3-only proteins, which binds extremely tightly to their target protein and induce potent cell killing in the relevant cell lines.

P19, line 441 “...all indicate that our CPs can effectively overcome clinical drug442 resistant BCL-2 mutations...”: Because the level of activity of the CP is not particularly high and have weak activity in cells, it is hard to see how this conclusion can be firmly proposed.

Table 1 and associated sensorgrams (and experimental section, where it is mentioned that the reproducibility “was confirmed by performing replicate determinations”: What does “replicate determinations” mean and why isn’t the data of these replicates included or at least SD or SEM presented. Ideally, all replicate sensorgrams should be shown. As such this table is not presenting a usable set of data, especially for values that are close to one another and upon which conclusions are drawn in the paper. Also, why is one concentration repeated twice in all SPR experiments? Is this the replicate that the authors are mentioning. If it is, that is not sufficient to provide statistical reliability of the data.

Figure 5:

- Panel b: the graph should not be cropped
- Panel c: this data is presented with no SD or SEM. No conclusions can be drawn from this data set.
- Panels, f,g,h: these should be all presented with the same scale on the y-axis. Why isn’t there any error bars on this graph? Was this experiment performed in several independent replicates?

Extended Figure 6:

- Panel a: missing statistical information

- Panels k,l,m,n: why aren't there error bars on all the curves in these panels? It is very unlikely that the reproducibility of the experiment is such that independent replicates were almost perfectly identical.

Minor issues

Page 11, line 242 "...but venetoclax binding seems to be more determinate.": the meaning of this part of the sentence is not clear. What do the authors mean by this?

Supplementary Figure 2a, data for CP2-6G against BCL2: discrepancy between the K_d value on the left-hand side of this graph and the K_D value on the steady state curve.

Reviewer #2 (Remarks to the Author):

Li et al have described the discovery of cyclic peptides that bind the pro-survival proteins BCL-2 and BCL-XL, along with their structural characterisation. The cyclic peptides reported here display different binding modes to published small molecule inhibitors, and thus are an interesting addition to the field, with the potential to form the basis for next generation inhibitors that can bypass resistance mutants. The identification of the non-conserved residue D111 in BCL2/A104 in BCL-XL is also a potentially useful observation for the design of high selectivity inhibitors. However, I did find that the authors oversold the selectivity of their cyclic peptides. A twofold difference in K_D values is hardly selective, at most it could be termed a slight preference. Particularly the claim in line 465 that cp3 and cp5 "selectively" bind BCL-XL, when they show only 2.5 & 2.1-fold difference, respectively. I also felt the effect of the cyclic peptides on clinical mutants like G101V was oversold, in line 441 the authors state their cyclic peptides "effectively overcome" clinical drug resistant BCL-2 mutations, but limits of cell permeability and lower affinity mean that even with a TAT-sequence, the IC₅₀ value for cp3 against BCL2-WT expressing cells is still much worse than venetoclax against G101V overexpressing cells. The authors should be careful to be realistic about the results of their studies. Cyclic peptides may be a new way to approach addressing the problem of clinical resistance, and while the novel binding mode and similar affinities for WT & G101V is a step forward, both the low affinity and the cell permeability would have to be addressed before claiming to "effectively overcome" clinical resistance.

Some specific points:

There were a significant number of typographical errors throughout the paper, along with some incorrect words (stimulation/simulation, malignancy in the abstract should be malignant, etc.).

Line 240 – this should read "consistent with" not "in consistent with"

Line 528: "discrepant" – maybe non-conserved is better?

In Figure 1 or 2, I thought a panel with an overlay of the cyclic peptide bound to BCL2/XL with a BH3 peptide would be a useful addition. I found myself wondering how the binding surface between the

cognate ligands and the cyclic peptides varied and it was difficult to visualise from the current figures. In a similar vein, in figure 3f, cp1 is overlaid with venetoclax, which was useful to see differences. However, I wondered if an additional view of this overlay from a similar perspective as 3a/b would be useful to get an appreciation for the differences in the breadth of interactions of the cyclic peptides compared with venetoclax.

The Rfree for the crystal structure 7Y90 seemed unusually high for a sub-2 Å structure, particularly with a Rmerge of 10%, and all other statistics of the data looking reasonable. Do the authors know why this is the case? It also has a very high clash score (20) for a high resolution structure. This may suggest the structure would benefit from further refinement and more careful inspection to minimise clashes and improve the Rfree.

The structure 7Y99 has 40% of the protein as unmodelled – this feels quite high to still be giving a good R/Rfree. Perhaps there is an error in the deposited sequence? The authors may want to double check this.

I do not have any experience in chi angle analysis, but I was a bit surprised in Fig. 3h to see that the G101V mutation seems to shift the chi1 distribution in the MD simulations for both F104 & F112 to the 3-radian peak, but the experimental structure has it at peak 1 (-1.5 radian). I felt like this needed comment in the description as to why the structure might have a less populated chi angle.

In Figure 5, I think E14 cells should be included in Fig 5a to show the relative BCL2/BCL-XL amounts, given the IC50 values are reported in c.

The IC50 of cp3-TAT in Daudi cells was 7.94 µM, which is comparable to that of venetoclax 9.12 µM and ABT-263 13.5 µM and S55746 10 µM. The authors describe this as a lower IC50 – but is it really significantly lower than the small molecule inhibitors, or just comparable? The justification for the better cellular inhibition is suggested to be “due to CPs’ special binding mode”, but I find this difficult to believe. It seems unlikely as the cause for better cellular activity. Perhaps the cyclic peptides have off-target binding in this cell line? Potentially other pro-survival proteins, or completely off-target. This seems a more likely explanation than a special binding mode.

I found myself wondering whether the A104/D111 “switch” influences the BCL2/XL selectivity of venetoclax? Does venetoclax (or other published inhibitors) interact with this residue?

“NO” in Fig. 5C – was this not determined? No inhibition?

Reviewer #3 (Remarks to the Author):

The authors developed a series of cyclic peptides that discriminate between BCL2 and BCLxL. the study

builds on a prior screen against a peptide library and IC50 data seem promising albeit higher than venetoclax. The advantage appears to lie in the activity against venetoclax resistant mutations in BCL2. I cannot comment on the structural data and the in vitro assessment of the tat-labeled peptides seems sufficiently solid. The authors might want to add in vivo evidence in a xenograft system including a comparison to venetoclax to provide a realistic assessment of single agent and/or combination activity. Please also clarify the frequency of the BCL2 mutations in venetoclax studies. Overall, a solid study with somewhat incremental novelty that chiefly lies in the activity against mutant BCL2.

Reviewer #4 (Remarks to the Author):

The authors applied structure-based design and phage-based screening to discover novel cyclic peptides (CP) which can be used as a potential selective inhibitor to BCL-2 and BCL-XL, and revealed the possible mechanism from structural and functional viewpoint. Six crystal structures of BCL-2-CP or BCL-XL-CP complexes were solved in this study. The authors performed structural analysis and molecular dynamics simulations to provide possible evidence of the selective binding mechanism of CP to the BCL-2 family.

11. Overall, this study provides insights into how cyclic peptides can potentially regulate cell apoptosis and demonstrate more advantages than the known small molecule inhibitors. Compared with traditional small molecule inhibitors, the macrocyclic structure usually enables the CPs to have high heat and protease stability. Furthermore, size and volume may also allow them to easily engage the binding pockets of important receptors. A large amount of work is reported in this paper and represents a major contribution towards discovering novel inhibitors for the BCL-2 family. However, some parts lack robustness, coherence, and reasonable explanations, and the writing needs further refinement. The following points must be addressed prior to publication:

1. The abstract and the final paragraph of the Introduction section of the paper appear to be too broad and lack focus. It would be helpful if the authors provided a brief summary of the key findings or conclusions they obtained from the study.
2. To enhance the visual consistency of the figures, the authors should maintain a consistent color scheme for the BCL-2 and BCL-XL proteins across different figures in the manuscript. And use different colors for different proteins. For instance, in Figure 2, the authors used the white color for the cartoon representation of both proteins, which is only used for BCL-XL in Figure 1. However, in Figure 3, white color and lime colors were used for BCL-2 WT and BCL-2 G101V.
3. The authors should clarify which unbound state the authors are comparing the complex structures of BCL-2-CP1 and BCL-XL-CP1 to in the manuscript and further clarify the novelty of the binding mode (Line 171-173, 177-178). The authors compared both complex structures to the ligand-free BCL-XL structure and drew the conclusion that CP1 doesn't induce the conformational change of BCL-2/BCL-XL. However, for BCL-2-CP1, conformational changes could be observed in the two helices compared with its own unbound state (Figure 2a).
4. Figure 2 shows the comparison of the bound state of BCL-2/BCL-XL with different ligands (CP1, BAX, Venetoclax). Can authors quantitatively compare the difference across the binding pockets of different

complexes?

5. In lines 216-217, the authors claim that there are conformational changes of F112 and F104, and these changes may reduce the binding affinities of Venetoclax and S55746 to BCL-2 mutant G101V. However, Figures 3c, 3e, and 3f show that F104 doesn't have an obvious conformational change when Venetoclax and S55746 are bound. At the same time, in line 245, the authors claimed that the conformational changes of F104 and F112 were not observed. The authors should have a detailed check and clarify this conclusion clearly.

6. How about the comparison of the distribution of chi1 angles of BCL-2 F104 and F112 from WT and G101V mutant in the states of S55746 bound? Is it similar to Venetoclax bound? At least it should be used to compare with other ligands as a supplementary figure.

7. Analysis based on MD data is needed to provide comprehensive evidence of the molecular mechanism of Venetoclax-resistant G101V mutation. And could the authors provide more conformational change information on the BCL-2 WT or G101V after binding from the MD simulations?

8. The authors should provide a detailed explanation of how Figure 3j shows G101V mutation directly affects the side-chain status of residues D103, R139, and E152. And explain how you get the network by detailed MD simulation analysis or experimental analysis.

9. The authors should provide a clearer explanation of the function of the sixth residue in the CPs on selectivity and binding affinity. For example, the selectivity change is clear between CP1 with 6E and CP2 with 6G. However, when the authors generated CP4 by mutating 6E of CP2 to 6G, there is no clear difference in selectivity between CP2 and CP4. Besides, the author should also use a quantitative method to describe selectivity. In addition, in line 344, the reviewers wonder how these results can prove 'suitable interface' instead of 'single residue' that is important for both binding affinity and selectivity?

10. The authors constructed a phage-based library, and four residues of CP were mutated, so around 160,000 sequences should be included in the library. In the manuscript, the authors only mentioned that 6th residue plays an essential role in the binding and selectivity of CPs. We expect more trends could be summarized from this library. In addition to residue 6, do any other residues provide some information for affecting binding affinity or selectivity to BCL-2/BCL-XL? If not, please provide more evidence.

Reviewer #5 (Remarks to the Author):

In this work, Wu and coworkers report the structural characterization of phage display derived peptides as inhibitors of BCL-2 and BCL-XL. Using structural data on their complex with the proteins, the authors further optimized the peptide to obtain cyclic peptide variants with improved selectivity for either proteins. They also tested their activity in cancer cells observing micromolar activity against certain lymphoma cell lines, validating their function in cells, albeit their cellular activity is significantly reduced (cp to their in vitro nanomolar KD) due to limitation in cell permeability. Interestingly, activity was observed with a clinically relevant variant of BCL-2 which shows reduced sensitivity to small molecule drugs. Interesting insights related to the impact of peptide binding on the conformation of the proteins as well as key residues discriminating selectivity of the cyclic peptides for the two homologous proteins are discussed.

Overall this is an excellent work that encompasses an impressive amount of work. The research is well described and conclusions are supported by the presented data. Findings are novel and relevant in the context of targeting protein-protein interactions with cyclic peptides. As a result, this work is considered appropriate and a good fit for the audience of this journal. Below are recommendations for revisions and corrections prior to publication, all of which are of minor nature:

1) The discussion section is very long and at times it appears redundant, as it reiterates concepts described earlier in the manuscript. A more concise summary of key results and conclusions is recommended. For example the section “Peptides can be considered....BCL-2” could be removed as it reiterates concepts already discussed earlier. Similarly, the whole section about genKIC and “Peptide-to-Small Molecule” method, and cyclosporine appears somewhat unnecessary as it does not directly relate to the present work. If deemed necessary, a more concise mention of these approaches is recommended.

2) IC50 values for the cellular assays are reported with in the micromolar range with two decimal units (e.g. “as 27.54 μ M, 13.18 μ M and 28.14 μ M,...”). Given the error in these assays, it seems unrealistic that this level of accuracy can be achieved. It is recommended to round up the value to the unit (not decimal).

3) In some places, wording appears colloquial “Perhaps that’s why in...”; “Trying to figure out why cp2...”

4) When discussing the phage display method for initial discovery or optimization of the peptides, it may be appropriate to briefly cite related phage display platforms for the selection of cyclic peptides. E.g. Nat Chem Biol. 2009 Jul;5(7):502-7. doi: 10.1038/nchembio.184; ACS Cent Sci. 2020 Mar 25;6(3):368-381. doi: 10.1021/acscentsci.9b00927; Nat Chem Biol. 2021 Jul;17(7):806-816. doi: 10.1038/s41589-021-00788-5

We sincerely thank all the five reviewers for their careful evaluation, strong overall encouragement, and constructive guidance in improving our manuscript.

Reviewer #1 (Remarks to the Author):

General comments

This manuscript by Li et al describes the development of cyclic peptides that target BCL-2 and BCL-XL, two members of the pro-survival protein family. By extensively analysing their binding mode via structural biology and molecular dynamics, they design peptides with specificity for one or the other. The paper provides some rationale design towards peptides which retain affinity for mutant versions of BCL-2 that have been identified in the clinic after venetoclax treatment.

Venetoclax has completely changed how certain cancers are treated, and continues to be tested against an ever-increasing range of tumours, as single agent or in combination. It remains the only BH3 mimetic drug currently approved. Others, targeting either BCL-2, BCL-XL or MCL1 are being clinically tested. As with all anti-cancer drugs used in the real world, clinicians have started cataloguing mutations conferring resistance to venetoclax. This is a critical issue as patients relapsing due to these altered forms of BCL-2 have little therapeutic options available. Solutions to this problem, with small molecules equally targeting all forms of BCL-2 or specifically the mutant forms, would have great therapeutic impact as second line treatment.

The data presented by Li et al aims at providing a solution to this issue and would therefore be an important advance in the field. Unfortunately, the paper falls short in three main areas and wouldn't be publishable in its current form in Nature Communications:

- Firstly, while the observations presented by the authors are interesting in the context of the cyclic peptide designed in this study, they are highly context specific and most likely not applicable to other small molecules. This is a conclusion that the authors themselves arrive at in their conclusion/discussion (P19, line 474-475). The broad applicability of their results is therefore arguable.

Many thanks for the above comments. Indeed, we agree that the binding mechanism of CPs we revealed in this work may not be applied to all types of small-molecule inhibitors against BCL-2 family proteins. As pointed out in the "Discussion" section, the binding assay of S55746 suggests that the "switch" may not be decisive for traditional small molecules deeply inserting into P2 and P4 pockets (Table 1, Supplementary Fig. 2 and 14), but may be critical for certain inhibitors with a special binding mode, like CPs. Our findings may provide a theoretical basis for the discovery and design of selective ligands with new scaffolds that don't insert deeply into P2 or P4 pockets.

- The biological activities presented (both binding and cellular) are far from therapeutically relevant. The SPR data looks robust (although more about this in my third point), but is still at least one log away from what is observed with endogenous BH3 only proteins and validated BH3 mimetics (single digit nanomolar or even picomolar). Moreover, the differences in selectivity observed are in some cases small and in ranges of activity that would be difficult to translate into a therapeutic effect in vivo. In particular, the on-target mechanism of the peptides with activity with IC50 values above 40 uM is questionable.

It is true that our CPs' activities are still far from therapeutically relevant at current stage, and further structural modification is indispensable to enhance their inhibitory activities. However, we speculate that due to the novel binding mode of CPs (not inducing dramatic protein conformational changes or deep pockets upon binding), they may not require a picomolar binding strength to exert functional effects, as compared with the traditional small-molecule inhibitors.

As for the relatively high micromolar IC50 values of CPs, the major reason could be their low efficiency in membrane penetration, although their TAT-modification could increase the penetrating efficiency of CPs to some extent. Another reason could be the cell line-specific response, especially the potential variance in their expression levels of BCL-2 or BCL-XL proteins. According to the reviewer's suggestions, we have conducted more functional experiments using different cell lines including the venetoclax-sensitive RS4:11 cells (**Fig. 5b**, Lines 389-395 on Page 17).

- The data in some places is not convincing. The SPR experiments are well conducted but no replicates (while mentioned in the text) are shown or included in the statistical calculation of the KD values. Similarly, statistics are missing on key figures with cellular data which diminish the impact of the conclusions drawn by the authors (see specific comments for examples). The demonstration that the cyclic peptides are acting by triggering mechanism-based apoptosis is also not satisfactory. The peptides at concentration above 40 uM are provoking apoptotic cell death but the authors do not present any functional data proving that this apoptotic cell death is indeed due to specific interaction to BCL-2 or BCL-XL. There are plenty of cellular models available where various pro-survival proteins are deleted that can help with this (see more details in specific remarks below).

Thanks for these constructive suggestions. In this revision, we have shown all the data from three replicates of SPR experiments (**Supplementary Fig. 2**), and recalculated the statistics for the SPR binding assays (**Table 1**), as well as the cellular data (**Fig. 5b**, **Supplementary Fig. 12**). Moreover, we added functional experiments using RS4:11 cells (**Fig. 5b**).

Specific remarks

Page 4, line 72 "it is actually very difficult to design BCL-2-selective inhibitors": this is not quite correct. Apart from ABT-737, ABT-263 and related compounds, most other advanced and validated BH3-mimetics have remarkable selectivity for their respective targets. What is true however, is that it is difficult to design selectivity "de novo".

According to the reviewer's helpful comments, we have revised this sentence (Lines 71-73 on Page 4).

Page 8, line 176-178, comment on the novelty of binding mode: this is somewhat of an overstatement. The conformational changes observed with most validated BH3 mimetics is probably a key element in their very high affinities, and in some cases selectivity, for their target. Also, it is not clear whether the analysis provided by the authors includes peptido-mimetics compounds or other peptides derived from BH3-only proteins.

The induced protein conformational change is indeed likely a key element of traditional small-molecule

BCL-2 family inhibitors for their high affinities and selectivity, with venetoclax as a successful example. However, their insertion deeply into the P2 and P4 pockets may be associated with the emergence of clinical drug resistance mutations (G101V or D103Y). In this work, we focus on and compare various types of inhibitors that are accompanied by reported protein-inhibitor complex structures in the PDB database (**Fig. 2, Supplementary Fig. 6**). Those inhibitors without available complex structures are not included in our comparison. The corresponding description has been modified accordingly (Lines 183-186 on Page 9).

Page 9, line 201-202 "...BCL-2 G101V mutant was about 1.1 μ M (close to the 0.8 μ M for BCL-2 WT),...": While these two values are indeed close, comparison between them is difficult since SD or SEM are not included in Table 1. It is not clear whether this data was calculated from several independent experiments. In addition, in other places, the authors draw difference conclusions from relatively similar differences between values.

Following this constructive suggestion, we have now shown all the data from three replicates and recalculated statistics for the SPR binding assays (**Table 1, Supplementary Fig. 2**). The related descriptions throughout the whole text have also been updated.

Page 14, lines 306 to 312: These conclusions are true but the changes they are associated with are small in comparison to the binding affinities that would be measured for peptides derived from the BH3-only proteins, so it not clear that this the observation is important.

Compared with the peptides derived from BH3-only proteins, the binding affinities of cp2 or cp4 are indeed weak. The observation here highlights the potentially critical role of the 6th residue of CPs in mediating their interactions with proteins, and suggests the importance of the single-residue "switch" together with subsequent data. CPs of better binding affinities and selectivity to BCL-2 or BCL-X_L proteins could be designed and developed based on these observations. A modification on this paragraph has been made (Lines 315-320 on Page 14).

Page 14, lines 323-324: "...selectivity of CPs are not simply defined by the size of their interfaces to proteins, yet likely determined by the very nature of CP residues,..": this is an obvious statement that is true for any binder to any proteins.

According to this comment, we have removed the corresponding sentence.

Page 16, lines 370-372: The binding affinity of CP1-TAT and CP3-TAT should be measured to be sure these constructs still have the same binding as their parent compounds.

Following the reviewer's suggestion, we have measured the binding affinities of cp1-TAT, cp2-TAT and cp3-TAT. The results indicated that addition of a TAT sequence does not cause a decrease in the binding as compared with their parent CPs (**Supplementary Table 5, Supplementary Fig. 2i, j, i' and j'**, Lines 377-378 on Page 16).

Page 16, lines 373 "...roughly consistent with their in vitro binding affinities...": what do the authors mean by this statement?

We tried to mean that the binding affinities obtained by SPR for cp1 and cp3 are positively correlated with their cellular inhibitory activities. These results could also serve as the secondary evidence to support the on-target function of CPs. The corresponding description has been modified in our manuscript (Lines 381-382 on Page 17).

Page 17, line 378 "...was even lower than those of the approved drug venetoclax...": this data (IC50 values close) is not presented with any SD or SEM. Comparison between these values is not possible, especially to draw such strong conclusion. Overall, these are fairly similar values.

Thanks again for pointing this out. Recalculated SD values are now presented for all the IC50 values. Indeed, the IC50 values of CP3-TAT, venetoclax, ABT-263 and S55746 are in a similar range for Daudi cells. Accordingly, we have described the results in a more reasonable way (Lines 387-389, Page 17).

Page 17, line 381 "...they may exhibit potent inhibitory functions as small-molecule drugs..." and P19, line 449: the data presented is not in line with this statement. In particular, the authors haven't chosen cell lines sensitive to BCL-2 or BCL-XL inhibition to start with: e.g. RS4;11 for BCL-2 selective compounds where inhibitors have low nanomolar inhibitory activity. These established cell lines with known sensitivity, combined with validated compounds should be used to assess the actual level of activity of the CPs. The data presented (~ 1 uM) is not suggestive of "potent inhibitory functions". In addition, the low cell membrane permeability is not sufficient to explain this low activity since the small molecules BH3 mimetic compounds are not particularly potent either.

Following this constructive suggestion of reviewer, we have measured the IC50 values of CPs and small-molecule BH3 mimetic compounds in the sensitive RS4;11 cell line (**Fig. 5b**). These new results showed that the small-molecule compounds inhibit cells at low nanomolar levels, verifying the effectiveness of our assay. However, the cellular inhibitory activities of CPs are still much lower (IC50 values much higher than corresponding KD values measured by SPR). We speculate that the low cell membrane permeability maybe the key reason why our CPs show such low inhibitory activities in different cell lines. The corresponding description has been modified in our manuscript (Lines 389-395 on Page 17).

P17, line 392-393: similar comment here. These IC50 values are well above what would be expected for on-target activity in cells dependent on BCL-2 for example.

Many thanks for the comments again. We have measured the IC50 values of CPs and BH3 mimetic compounds in the RS4;11 cell line (**Fig. 5b**). The results from four different cell lines indicate that venetoclax and ABT-263 possess clear on-target activity and cell-selectivity (Lines 389-393 on Page 17).

P17, line 395: Similar comment and at that level of activity, it is possible that the killing observed is not due to selective BCL-2 inhibition.

To verify the selective BCL-2 inhibition, we have measured the IC50 values of CPs and BH3 mimetic compounds in RS4;11 cell line (**Fig. 5b**). Corresponding descriptions in the manuscript have also

been updated (Lines 413-415 on Page 18).

P17, line 399 and P18, line 418-419 "...another hallmark detecting method for apoptosis,...": while these are classical phenotypic hallmarks of apoptosis, there are not providing information demonstrating on target/mechanism based activity. For lists of criteria for mechanism-based activity, refer to the following reviews: Lessene et al, Nat Rev Drug Discover, 2008; Soderquist et al, Mol Canc Ther, 2016 and Villalobos Ortiz et al, Cell Death Diff, 2018.

Following the reviewer's advice, we tried to test the on-target effects of CPs in Jurkat cells by monitoring the amount of mitochondrial Cytochrome C (cytC), in the absence or presence of the BAX-inhibiting peptide (BIP-V5). After a lot of efforts to build up a reliable assaying system, we found that BIP-V5 indeed could block the cytC release induced by venetoclax. Unfortunately, we could only see a similar trendy for TAT-tagged CPs, but could not achieve a very concrete conclusion, given their weak cellular activities (largely due to their low membrane permeability) and the relatively limited time for revision. However, we think the co-crystal structures and SPR results, showing the interactions and nanomolar-level binding affinities between CPs and BCL-2/BCL-X_L proteins, would at least to a certain extent, support the on-target mechanism of our CPs. Related descriptions have also been updated in the revised manuscript (Lines 399-340 on Page 17, and Lines 416 -418 on Page 18).

P19, line 437 "...onto the flat surface of the hydrophobic groove, without inducing conformational changes of the $\alpha 3$ and $\alpha 4$ helices...": considering the relatively weak level of affinity and very weak cellular activity, it is possible that the lack of conformational changes to accommodate these peptides is also a drawback. The true comparison here would be with peptides derived the BH3-only proteins, which binds extremely tightly to their target protein and induce potent cell kicking in the relevant cell lines.

We agree that lack of conformational changes and deep insertion into the binding pockets of BCL-2 and BCL-X_L proteins, could be one of the reasons for our CPs to exhibit relatively lower binding affinities than those traditional small-molecule inhibitors and peptides derived from BH3-only proteins. However, this unique binding mode of CPs also provides new ideas. For example, the single-residue "switch" (BCL-2/D111 and BCL-X_L/A104) we identified here, may inspire the development of novel inhibitors (including further modified CPs) with a better target selectivity or a stronger ability to override drug-resistant mutations. The large molecular size of CPs renders them a great potential to increase their binding affinities to target proteins by suitable modifications, yet also a challenge to achieve better cell-membrane permeability.

P19, line 441 "...all indicate that our CPs can effectively overcome clinical drug resistant BCL-2 mutations...": Because the level of activity of the CP is not particularly high and have weak activity in cells, it is hard to see how this conclusion can be firmly proposed.

We have modified this sentence according to the reviewer's comments (Lines 413-415 on Page 18).

Table 1 and associated sensorgrams (and experimental section, where it is mentioned that the reproducibility "was confirmed by performing replicate determinations": What does "replicate determinations" mean and why isn't the data of these replicates included or at least SD or SEM presented. Ideally, all replicate sensorgrams

should be shown. As such this table is not presenting a usable set of data, especially for values that are close to one another and upon which conclusions are drawn in the paper. Also, why is one concentration repeated twice in all SPR experiments? Is this the replicate that the authors are mentioning. If it is, that is not sufficient to provide statistical reliability of the data.

Many thanks for the comments and suggestions. In SPR experiments, we tested one concentration of samples twice in order to verify the stability of this assaying system. We have now shown all the sensorgrams from independent replicates for SPR experiments and recalculated the statistics in the revised manuscript (**Table 1, Supplementary Fig. 2**).

Figure 5:

- Panel b: the graph should not be cropped
- Panel c: this data is presented with no SD or SEM. No conclusions can be drawn from this data set.
- Panels, f,g,h: these should be all presented with the same scale on the y-axis. Why isn't there any error bars on this graph? Was this experiment performed in several independent replicates?

Based on reviewer's comments, we have modified these panels in the new **Fig. 5 and Supplementary Fig. 11**. For the last three panels (f, g, h), the data were from six independent replicates.

Extended Figure 6:

- Panel a: missing statistical information
- Panels k,l,m,n: why aren't there error bars on all the curves in these panels? It is very unlikely that the reproducibility of the experiment is such that independent replicates were almost perfectly identical.

Thanks for pointing this out. Data for these panels were from 6 independent replicates. This figure has been reorganized and is presented as **Supplementary Fig. 11** now in the revised manuscript.

Minor issues

Page 11, line 242 "...but venetoclax binding seems to be more determinate.": the meaning of this part of the sentence is not clear. What do the authors mean by this?

We meant to say that the conformational changes of F104 and F112 caused by G101V mutation, could be restrained or reversed by binding of venetoclax. We have modified this sentence in the manuscript (Lines 242-244 on Page 11).

Supplementary Figure 2a, data for CP2-6G against BCL2: discrepancy between the K_d value on the left-hand side of this graph and the K_D value on the steady state curve.

We apologize for this mistake. Annotation on this figure has been corrected in the revised manuscript.

Reviewer #2 (Remarks to the Author):

Li et al have described the discovery of cyclic peptides that bind the pro-survival proteins BCL-2 and BCL-XL, along with their structural characterisation. The cyclic peptides reported here display different binding modes to published small molecule inhibitors, and thus are an interesting addition to the field, with the potential to form the basis for next generation inhibitors that can bypass resistance mutants. The identification of the non-conserved residue D111 in BCL2/A104 in BCL-XL is also a potentially useful observation for the design of high selectivity inhibitors. However, I did find that the authors oversold the selectivity of their cyclic peptides. A twofold difference in KD values is hardly selective, at most it could be termed a slight preference. Particularly the claim in line 465 that cp3 and cp5 “selectively” bind BCL-XL, when they show only 2.5 & 2.1-fold difference, respectively. I also felt the effect of the cyclic peptides on clinical mutants like G101V was oversold, in line 441 the authors state their cyclic peptides “effectively overcome” clinical drug resistant BCL-2 mutations, but limits of cell permeability and lower affinity mean that even with a TAT-sequence, the IC50 value for cp3 against BCL2-WT expressing cells is still much worse than venetoclax against G101V overexpressing cells. The authors should be careful to be realistic about the results of their studies. Cyclic peptides may be a new way to approach addressing the problem of clinical resistance, and while the novel binding mode and similar affinities for WT & G101V is a step forward, both the low affinity and the cell permeability would have to be addressed before claiming to “effectively overcome” clinical resistance.

We greatly appreciate the reviewer's critical comments that accurately identify both the strengths and weaknesses of our work. Indeed, the selectivity of cp3 and cp5 is much weaker than that of cp1. We wanted to note that modification of single CP residues could cause a clear change in their selectivity. Based on this, we propose two possible mechanisms that affect the selectivity of CPs (Lines 460-462 on Page 20). The binding affinities to BCL-X_L of cp3 and cp5 increased by about 130- and 300-fold than that of cp1, respectively. In our future studies, we will continue to modify and optimize the CPs to enhance their binding affinity and selectivity, building upon the mechanisms proposed above.

Some specific points:

There were a significant number of typographical errors throughout the paper, along with some incorrect words (stimulation/simulation, malignancy in the abstract should be malignant, etc.).

Line 240 – this should read “consistent with” not “in consistent with”

Line 528: “discrepant” – maybe non-conserved is better?

Many thanks for pointing these mistakes out. We have corrected them accordingly.

In Figure 1 or 2, I thought a panel with an overlay of the cyclic peptide bound to BCL2/XL with a BH3 peptide would be a useful addition. I found myself wondering how the binding surface between the cognate ligands and the cyclic peptides varied and it was difficult to visualise from the current figures. In a similar vein, in figure 3f, cp1 is overlaid with venetoclax, which was useful to see differences. However, I wondered if an additional view of this overlay from a similar perspective as 3a/b would be useful to get an appreciation for the differences in the breadth of interactions of the cyclic peptides compared with venetoclax.

Thank you for your advice. In **Fig. 2d, h** we added a panel about BCL-2/BCL-X_L complex with CP1

and now can clearly see the surface differences between BCL-2 BAX/BCL-X_L BID. We have added an additional view of **Fig. 3g** accordingly.

The R_{free} for the crystal structure 7Y90 seemed unusually high for a sub-2 Å structure, particularly with a R_{merge} of 10%, and all other statistics of the data looking reasonable. Do the authors know why this is the case? It also has a very high clash score (20) for a high resolution structure. This may suggest the structure would benefit from further refinement and more careful inspection to minimise clashes and improve the R_{free}.

In accordance with the reviewer's suggestion, we thoroughly reviewed the raw data of PDB 7Y90 (cp1-bound BCL-2). It appears that the quality of the diffraction pattern is relatively poor, with some diffraction points being split. Subsequently, we reprocessed the data and refined the structure once more. As a result, the clash score significantly decreased to 6. However, we regret to find that there was no noticeable improvement in the R factor/R_{free} values. We have contacted the PDB database and updated this deposition.

The structure 7Y99 has 40% of the protein as unmodelled – this feels quite high to still be giving a good R/R_{free}. Perhaps there is an error in the deposited sequence? The authors may want to double check this.

For the ease of protein expression and crystallization, we have removed residues 27-81 and the C-terminus of BCL-X_L. Unfortunately, there was an oversight and we accidentally uploaded the full-length sequence during deposition. We sincerely apologize for this mistake and have rectified the issue by re-uploading the correct protein sequence to PDB.

I do not have any experience in chi angle analysis, but I was a bit surprised in Fig. 3h to see that the G101V mutation seems to shift the chi1 distribution in the MD simulations for both F104 & F112 to the 3-radian peak, but the experimental structure has it at peak 1 (-1.5 radian). I felt like this needed comment in the description as to why the structure might have a less populated chi angle.

Figure R1. Free energy of F104 and F112 along the chi1 angle. For continuity of display, we shifted the left side (angles less than 2.0 radian) to the right side (+2 π). The free energy is calculated by $-RT\ln(\rho)$, RT is the product of the molar gas constant, R , and the temperature, T . ρ is the probability distribution of different chi1 obtained from the statistics of the simulated trajectory.

This is a very good question. We also noticed this phenomenon and tried to investigate the reason carefully. The experimental data indicates that in BCL-2 G101V-cp1 complex structure, the -1.5-radian chi1 is the dominant conformation. However, both states (chi1 angle at the 3-radian or -1.5-radian) are allowed in the simulations; yet the main peak of chi1 seems at 3-radian, inconsistent with the experimental data (**Fig. 3i**). This change in chi1 distribution occurs after the G101V mutation. Compared to Gly, Val has a larger side chain, which can form complex interactions with E152 and F112, subsequently inducing a series of chi angle changes. We believe this discrepancy arises from the limitations of the force field, which is not accurate enough to describe the complex, subtle, and somewhat multi-body interactions mentioned above. The energy difference between the -1.5-radian and 3-radian state is small, only 1.1-1.3 kcal/mol (**Figure R1**). Correcting such errors is an extremely challenging and systematic task, which deviates from the main goal of this work. What we are trying to emphasize here is that the -1.5-radian chi1 state is allowed after binding to the cp1, but not allowed after binding to the venetoclax, which indicates that the latter exerts additional force on the mutated side chains. We believe that this qualitative conclusion is reliable.

In Figure 5, I think E14 cells should be included in Fig 5a to show the relative BCL2/BCL-XL amounts, given the IC50 values are reported in c.

Following this suggestion, we have added the WB results of BCL-2/BCL-X_L amounts in E14 cells (**Supplementary Fig. 11c**), and we also added the WB results in RS4:11 cells (**Supplementary Fig. 11d**, Lines 363 to 366 on Page 16).

The IC50 of cp3-TAT in Daudi cells was 7.94 μ M, which is comparable to that of venetoclax 9.12 μ M and ABT-263 13.5 μ M and S55746 10 μ M. The authors describe this as a lower IC50 – but is it really significantly lower than the small molecule inhibitors, or just comparable? The justification for the better cellular inhibition is suggested to be “due to CPs’ special binding mode”, but I find this difficult to believe. It seems unlikely as the cause for better cellular activity. Perhaps the cyclic peptides have off-target binding in this cell line? Potentially other pro-survival proteins, or completely off-target. This seems a more likely explanation than a special binding mode.

The IC50 value of cp3-TAT in Daudi cells is indeed comparable to those of small-molecule inhibitors. Considering the very low membrane penetration efficiency of CPs, these data at least seem encouraging. One of the well-known advantages of cyclic peptides is their high selectivity and low toxicity, rendering them a less chance of being off-targeted. We also tested the binding affinities of cp3 to all pro-survival proteins, and the results showed that it only has a high binding affinity to BCL-2, BCL-X_L and BCL-w (**Table 1**).

I found myself wondering whether the A104/D111 “switch” influences the BCL2/XL selectivity of venetoclax? Does venetoclax (or other published inhibitors) interact with this residue?

Figure R2. Key residue analysis of BCL-2 within 5 Å around venetoclax.

After analyzing the complex structure of venetoclax and BCL-2 (**Figure R2**), we found the shortest distance of D111 to venetoclax is 4.3 Å, and there is no other direct interaction between D111 and venetoclax. Meanwhile, we found S55746 can interact with BCL-2 D111 directly, but D111 seems not the key reason influencing its selectivity (Lines 467-475, on page 20, **Table 1** and **Supplementary Fig. 14**).

“NO” in Fig. 5C – was this not determined? No inhibition?

“NO” here means no inhibition. We have added a note in the legends of **Fig. 5**.

Reviewer #3 (Remarks to the Author):

The authors developed a series of cyclic peptides that discriminate between BCL2 and BCLxL. the study builds on a prior screen against a peptide library and IC50 data seem promising albeit higher than venetoclax. The advantage appears to lie in the activity against venetoclax resistant mutations in BCL2. I cannot comment on the structural data and the in vitro assessment of the tat-labeled peptides seems sufficiently solid. the authors might want to add in vivo evidence in a xenograft system including a comparison to venetoclax to provide a realistic assessment of single agent and/or combination activity. Please also clarify the frequency of the BCL2 mutations in venetoclax studies. Overall, a solid study with somewhat incremental novelty that chiefly lies in the activity against mutant BCL2.

Many thanks for the positive comments on our study and for the suggestion about *in vivo* assessments in a xenograft system. Regrettably, due to time constraints during the revision period, we were unable to conduct such experiments. This is primarily because preparing a large quantity of CPs is required, given their current limited cellular inhibitory activities. Moving forward, our future research will focus on optimizing the structure and modification of CPs to enhance their activities, particularly their membrane permeability. This will facilitate more practical implementation of *in vivo* experiments.

Regarding the frequency of BCL-2 mutations, previous clinical studies have indicated the presence of the G101V mutation in cases of CLL-type progression during venetoclax treatment. Specifically, this mutation was detected in 7 out of 15 clinical cases of this nature (Cancer Discovery, 2019 Mar;9(3):342-353. doi: 10.1158/2159-8290.).

Reviewer #4 (Remarks to the Author):

The authors applied structure-based design and phage-based screening to discover novel cyclic peptides (CP) which can be used as a potential selective inhibitor to BCL-2 and BCL-XL, and revealed the possible mechanism from structural and functional viewpoint. Six crystal structures of BCL-2-CP or BCL-XL-CP complexes were solved in this study. The authors performed structural analysis and molecular dynamics simulations to provide possible evidence of the selective binding mechanism of CP to the BCL-2 family. Overall, this study provides insights into how cyclic peptides can potentially regulate cell apoptosis and demonstrate more advantages than the known small molecule inhibitors. Compared with traditional small molecule inhibitors, the macrocyclic structure usually enables the CPs to have high heat and protease stability. Furthermore, size and volume may also allow them to easily engage the binding pockets of important receptors. A large amount of work is reported in this paper and represents a major contribution towards discovering novel inhibitors for the BCL-2 family. However, some parts lack robustness, coherence, and reasonable explanations, and the writing needs further refinement. The following points must be addressed prior to publication:

1. The abstract and the final paragraph of the Introduction section of the paper appear to be too broad and lack focus. It would be helpful if the authors provided a brief summary of the key findings or conclusions they obtained from the study.

We are grateful for the above constructive comments. We have revised the abstract and the final paragraph of the Introduction section accordingly.

2. To enhance the visual consistency of the figures, the authors should maintain a consistent color scheme for the BCL-2 and BCL-XL proteins across different figures in the manuscript. And use different colors for different proteins. For instance, in Figure 2, the authors used the white color for the cartoon representation of both proteins, which is only used for BCL-XL in Figure 1. However, in Figure 3, white color and lime colors were used for BCL-2 WT and BCL-2 G101V.

In response to this valuable suggestion provided by the reviewer, we have adjusted the coloring of these figures (**Figs. 2-4, Supplementary Figs. 3, 5, 6, 8**).

3. The authors should clarify which unbound state the authors are comparing the complex structures of BCL-2-CP1 and BCL-XL-CP1 to in the manuscript and further clarify the novelty of the binding mode (Line 171-173, 177-178). The authors compared both complex structures to the ligand-free BCL-XL structure and drew the conclusion that CP1 doesn't induce the conformational change of BCL-2/BCL-XL. However, for BCL-2-CP1, conformational changes could be observed in the two helices compared with its own unbound state (Figure 2a).

Thanks for pointing this out. The main reason we used the unbound structure of BCL-X_L to compare with both cp-1 bound BCL-X_L and BCL-2 is that those structures were all obtained by crystallography, while the only available unbound structure of BCL-2 was solved using NMR. We found that the NMR structure of BCL-2, which was calculated as its average conformation in solution, lied in a middle state between the small-molecule ligand-bound state and the cp1-bound state (**Fig. 2a**). These three different conformational states of BCL-2 imply the potential dynamic status of this protein. Interestingly, both CP1-bound BCL-2 and BCL-X_L structures exhibit the same conformation as the crystal structure

of unbound BCL-X_L. All above results suggest the novelty of CPs' binding mode. In addition, another interesting point is that our CPs do not need to produce P2 and P4 pockets in BCL-X_L or BCL-2 proteins for their binding, which is different from other types of inhibitors that we currently know. The corresponding descriptions within the manuscript have been modified (Lines 173 to 175 on Page 8).

4. Figure 2 shows the comparison of the bound state of BCL-2/BCL-XL with different ligands (CP1, BAX, Venetoclax). Can authors quantitatively compare the difference across the binding pockets of different complexes?

According to this very helpful suggestion, we quantitatively measured the solvent-accessible surface area (SASA) changes of certain key residues on the BH3-only binding surface in different complex states of BCL-2 and BCL-X_L. The results are consistent with our structural analysis that cp1 targets BCL-2/BCL-X_L in a novel binding mode that it does not need to induce conformational changes of α 3 and α 4 helices to unveil the deep pockets (such as P2 and P4) (Lines 175-180 on Page 8, **Supplementary Fig. 7**).

5. In lines 216-217, the authors claim that there are conformational changes of F112 and F104, and these changes may reduce the binding affinities of Venetoclax and S55746 to BCL-2 mutant G101V. However, Figures 3c, 3e, and 3f show that F104 doesn't have an obvious conformational change when Venetoclax and S55746 are bound. At the same time, in line 245, the authors claimed that the conformational changes of F104 and F112 were not observed. The authors should have a detailed check and clarify this conclusion clearly.

The conformational changes of F112 and F104 potentially caused by G101V mutation, is one of the interesting findings we unveiled in this study. First, we found that the conformations of F112 and F104 are different in the cp1-bound structures of BCL-2 WT and G101V mutant. Then we utilized the MD simulation, which indicated that these conformational changes are caused by G101V mutation, but not the binding of cp1. Meanwhile, the MD simulation results also indicated that Venetoclax binding could influence the chi angle distribution of F104 and F112, likely restraining their conformational changes caused by G101V mutation. This could be the main reason why there are no obvious conformational changes of these two residues in the venetoclax-bound structures of BCL-2 G101V mutant as compared with the WT. All above observations indicate that BCL-2 G101V mutation may not only influence the conformation of E152, but also influence that of F104 and F112 consequently. The corresponding description and conclusion within the manuscript have been modified (Lines 237-255 on Page 11).

6. How about the comparison of the distribution of chi1 angles of BCL-2 F104 and F112 from WT and G101V mutant in the states of S55746 bound? Is it similar to Venetoclax bound? At least it should be used to compare with other ligands as a supplementary figure.

Following the reviewer's suggestion, we have added the simulation data about the chi1 angle changes of BCL-2 F104 and F112 from WT and G101V mutant in the states of S55746 bound. The results indicated that S55746, like Venetoclax, could influence the chi angles distribution of F104 and F112 (Lines 250-253 on Page 11, **Fig. 3k**).

7. Analysis based on MD data is needed to provide comprehensive evidence of the molecular mechanism of Venetoclax-resistant G101V mutation. And could the authors provide more conformational change information on the BCL-2 WT or G101V after binding from the MD simulations?

Figure R3. MD analysis on the conformational changes of key BCL-2 residues. Panels a-e are from the crystal structures and f-h are from the simulation. The red arrows indicate the relative conformation of F104 and F112.

(1) A conformation comparison of key BCL-2 residues (G/V101, F104, F112 and E152) in the complex crystal structures (**Figure R3a, b, d, e**) showed that no matter binding to venetoclax or cp1, the BCL-2 G101V mutation causes an orientation change for the side chain of E152, whose chi1 changes from around -80° to around 180° (as indicated in the parentheses near the residues). We also found that

the aromatic heterocycle on venetoclax hinders the twisting of the chi1 angle of F104 (from 180° to -80°), while cp1 allows such a side-chain conformation change of F104 (enlarged view in **Panel c**). In the simulation, the relative posture and orientation of the F104 side chain and the heterocycle of venetoclax are almost fixed. **Figure R3f** shows the change of the minimum distance between the F104 side chain and the heterocycle functional group within the simulation time. No matter in the case of WT or G101V, this distance is stable at 3.5 Å to 4.0 Å.

(2) Based on the MD simulation results of G101V mutant, we found that after binding of venetoclax, the chi1 of F104 is indeed locked at around 180°, while the side chain of F112 is allowed to rotate (**Figure R3g**). However, after binding of cp1, the side chains of both F104 and F112 are allowed to rotate (**Figure R3h**). These results from simulations confirmed our deduction on account of the crystal structure, that venetoclax binding locks the side-chain conformation of F104. Furthermore, in the simulations there is a clear correlation between the conformational changes of the side chains of F104, F112 and E152, no matter cp1 is bound or not. However, this correlation is greatly suppressed when venetoclax is bound (**Table R1**).

Table R1. The conformation changes correlation of these key amino acids.

	substrate-free			venetoclax			cp1				
	F104	F112	E152	F104	F112	E152	F104	F112	E152		
F104	1.00	0.37	-0.10	F104	1.00	0.01	-0.17	F104	1.00	0.67	-0.39
F112	0.37	1.00	-0.49	F112	0.01	1.00	-0.13	F112	0.67	1.00	-0.32
E152	-0.10	-0.49	1.00	E152	-0.17	-0.13	1.00	E152	-0.39	-0.32	1.00

8. The authors should provide a detailed explanation of how Figure 3j shows G101V mutation directly affects the side-chain status of residues D103, R139, and E152. And explain how you get the network by detailed MD simulation analysis or experimental analysis.

Figure R4. Positional relationship between ligands, BCL-2 D103/R139 and other key residues.

(1) The association between E152 conformation and G101V mutation has been explained clearly in previous papers (Nature communications 10, 2385 (2019)) and in our work. Here is a brief summary: G101V mutation causes a conformational change in the side chain of nearby residue E152, which is an important participant for the formation of classic ligand binding pocket on BCL-2, consequently triggering possible conformational changes of residues F112 and F104.

(2) According to the crystal structures of venetoclax-bound BCL-2 WT and G101V mutant (**Figure R4a, b**), the states of D103 and R139 are not changed in the G101V mutant as compared with the WT. R139 is actually located at the far end and does not directly participate in the venetoclax binding. While in the cp1-bound crystal structures, R139 is far away from the G101V mutation site, yet near to cp1 (**Figure R4c, d**). The effect of G101V mutation on R139 residue may be mediated by the overall conformational change of cp1. The terminal carboxyl group of the side chain of D103 is affected by the conformational change of F104. When the chi angle of F104 is around -80 degree (**Figure R4d**), it may form a hydrogen bond interaction with the carboxyl group on D103, triggering the change of chi2 of the latter.

In summary, the association between D103 or R139 and BCL-2 G101V mutant mainly occurs when binding to cp1, and both are through long-range conduction. However, venetoclax binding may block the conformational change of D103, consistent with its effects on F104 and F112.

The network is a simple abstraction of the correlation graph (**Figure 3j**), in order to present the relationship between important residues more clearly. If there is a strong correlation between the state changes of two residues, it will be connected by a thick arrow; and if there is a weak correlation between the state changes of two residues, it will be connected by a thin arrow. The detailed methods have been added to the molecule dynamics (MD) simulation method section (Lines 665-667 on Page 35).

9. The authors should provide a clearer explanation of the function of the sixth residue in the CPs on selectivity and binding affinity. For example, the selectivity change is clear between CP1 with 6G and CP2 with 6E. However, when the authors generated CP4 by mutating 6E of CP2 to 6G, there is no clear difference in selectivity between CP2 and CP4. Besides, the author should also use a quantitative method to describe selectivity. In addition, in line 344, the reviewers wonder how these results can prove ‘suitable interface’ instead of ‘single residue’ that is important for both binding affinity and selectivity?

Table R2. Binding affinity and selectivity changes of CPs.

	cp1	cp2	cp3	cp4	cp5
BCL-2	0.65 μ M	0.20 μ M	0.24 μ M	0.08 μ M	0.08 μ M
BCL-X_L	>12.14 μ M	0.73 μ M	0.09 μ M	0.47 μ M	0.04 μ M
Selectivity to BCL-2	>19 fold	3.6 fold	-2.7 fold	5.9 fold	-2 fold

Figure R5. Structures of CPs. The red shade indicates the residues not directly involved in the interactions to proteins.

The function of the 6th residue in the CPs on selectivity and binding affinity is indeed an interesting question yet difficult to be fully answered. From the figure above, it can be seen that all of our CPs are composed of 11 residues + linker (cADT). Only residues at positions 3, 9, 10, and the 6th are non-conserved. The differences in affinity and selectivity between CPs are theoretically mainly determined by these non-conserved residues. From the analysis of interactions between individual residues, those at positions 3, 9, and 10 do not directly interact with the target proteins (**Fig. 1**).

Structural analysis revealed that the 6th residue of cp1 is related to BCL-X_L/A104 and BCL-2/D111, which are the only non-conserved pair of residues on the binding surface. Combining co-crystal structure analysis and single-residue mutation assays, we found the single-residue discrepancy of BCL-2/D111 and BCL-X_L/A104 might be one of the key reasons for the selectivity of cp1, which is more than 19-fold selective to BCL-2 than BCL-X_L (**Table R2**).

From the BCL-2 targeted screening we obtained a new CP annotated as cp2. We found that the residue 6E in cp2 (**Figure R5**) might be the key reason for its improved binding affinity to both BCL-2 and BCL-X_L, while this increased binding affinity to BCL-X_L also led to its preference to BCL-2 dropped to about 3.6-fold (**Table R2**). These results also suggested that the 6th residue of CPs (as well as corresponding BCL-2/D111 and BCL-X_L/A104) may play key roles for the selectivity.

Interestingly, a new CP from the BCL-X_L targeted screening, cp3 with the 6P residue (**Figure R5**), exhibited a binding affinity to BCL-X_L of 0.09 μ M (**Table R2**) and reversed the preference to BCL-X_L for 2.7-fold (**Table R2**). Structural analysis suggested that the 6P of cp3 may form a “suitable interface” together with BCL-X_L A104 (**Figure. 4d-f**). Furthermore, the cp5, as a derivative of cp2 only with its 6E substituted by 6P (**Figure R5**), demonstrated enhanced binding affinity (0.04 μ M) and reversed preference to BCL-X_L (**Table R2**). These results indicated that this region of the protein-CP interface, formed by BCL-X_L A104 and 6P of CPs, may be very important for the binding affinity and selectivity. We originally used the phrase “suitable interface” to describe the fitness between the molecular

surfaces of BCL-X_L A104 and 6P. The corresponding description in the manuscript has been modified (Lines 349-351 on Page 15).

10. The authors constructed a phage-based library, and four residues of CP were mutated, so around 160,000 sequences should be included in the library. In the manuscript, the authors only mentioned that 6th residue plays an essential role in the binding and selectivity of CPs. We expect more trends could be summarized from this library. In addition to residue 6, do any other residues provide some information for affecting binding affinity or selectivity to BCL-2/BCL-XL? If not, please provide more evidence.

After the screening against BCL-2 and BCL-X_L, we randomly picked up 30 clones for sequencing. The obtained sequences were then analyzed, and the results are shown in **Fig. 4a** and **Supplementary Table 3**. We indeed observed another conserved residue that could be crucial for the protein binding. The 3rd residue is highly conserved in both selections, and the residues isoleucine and valine at this position might be important for the binding of both proteins (**Figure R6a-d**). The 9th and 10th residues are not conserved for both proteins, which might be less important for the protein binding compared with other residues in the CPs. The 6th proline residue is also highly conserved for the BCL-X_L binding, but not for BCL-2 (**Figure R6a, c**). All the above results suggest that the binding selectivity between two proteins could be largely determined by this 6th residue.

Figure R6. The second round of phage-based screening results. a. The phage library screened the monoclonal sequencing results for BCL-2. **b.** Sequence abundance analysis of BCL-2 Phage screening. **c.** The phage library screened the monoclonal sequencing results for Bcl-X_L. **d.** Sequence abundance analysis of BCL-XL Phage screening.

Reviewer #5 (Remarks to the Author):

In this work, Wu and coworkers report the structural characterization of phage display derived peptides as inhibitors of BCL-2 and BCL-XL. Using structural data on their complex with the proteins, the authors further optimized the peptide to obtain cyclic peptide variants with improved selectivity for either proteins. They also tested their activity in cancer cells observing micromolar activity against certain lymphoma cell lines, validating their function in cells, albeit their cellular activity is significantly reduced (cp to their in vitro nanomolar KD) due to limitation in cell permeability. Interestingly, activity was observed with a clinically relevant variant of BCL-2 which shows reduced sensitivity to small molecule drugs. Interesting insights related to the impact of peptide binding on the conformation of the proteins as well as key residues discriminating selectivity of the cyclic peptides for the two homologous proteins are discussed.

Overall this is an excellent work that encompasses an impressive amount of work. The research is well described and conclusions are supported by the presented data. Findings are novel and relevant in the context of targeting protein-protein interactions with cyclic peptides. As a result, this work is considered appropriate and a good fit for the audience of this journal. Below are recommendations for revisions and corrections prior to publication, all of which are of minor nature:

1) The discussion section is very long and at times it appears redundant, as it reiterates concepts described earlier in the manuscript. A more concise summary of key results and conclusions is recommended. For example the section “Peptides can be considered...BCL-2” could be removed as it reiterates concepts already discussed earlier. Similarly, the whole section about genKIC and “Peptide-to-Small Molecule” method, and cyclosporine appears somewhat unnecessary as it does not directly relate to the present work. If deemed necessary, a more concise mention of these approaches is recommended.

Many thanks for the constructive comments. Based on the reviewer’s suggestion, we have removed these redundant parts and reorganized sentences in the discussion section.

2) IC50 values for the cellular assays are reported with in the micromolar range with two decimal units (e.g. “as 27.54 μ M, 13.18 μ M and 28.14 μ M,...”). Given the error in these assays, it seems unrealistic that this level of accuracy can be achieved. It is recommended to round up the value to the unit (not decimal).

We appreciate the reviewer’s advice and have changed the data unit accordingly.

3) In some places, wording appears colloquial “Perhaps that’s why in...”; “Trying to figure out why cp2...”

Thanks again for point out this issue. We have adjusted these colloquial descriptions in our revised manuscript.

4) When discussing the phage display method for initial discovery or optimization of the peptides, it may be appropriate to briefly cite related phage display platforms for the selection of cyclic peptides. E.g. Nat Chem Biol. 2009 Jul;5(7):502-7. doi: 10.1038/nchembio.184; ACS Cent Sci. 2020 Mar 25;6(3):368-381. doi: 10.1021/acscentsci.9b00927; Nat Chem Biol. 2021 Jul;17(7):806-816. doi: 10.1038/s41589-021-00788-5

According to the reviewer's valuable suggestion, we have cited these representative and classic references in the manuscript (Line 273 on Page 12).

REVIEWER COMMENTS

Reviewer #1 (Remarks to the Author):

Overall, the authors have addressed a number of the concerns raised. In particular they have added repeats and a few critical experiments, especially in cells that are dependent on BCL-2 for survival, helping greatly with benchmarking the activity of these designed cyclic peptides vs small molecules.

There are however some remaining concerns that the additional data has not fully addressed:

The authors seem to have misinterpreted my point about the need to include data for RS4,11 cells: it wasn't to validate ABT-199's or ABT-263's on target activity, which is well documented. It was to benchmark the cyclic peptides in a validated cell line that is indeed known to be BCL-2 dependent. The result of this experiment are clearly indicating that the CPs do not efficiently induce BCL-2-dependent cell death and that the death observed at these high concentration, in all cell lines, is probably non-specific. Also, levels of pro-survival proteins do not dictate their dependence to these proteins. Dependencies is analysed using validated tool compounds such as ABT-199, A-1331852 or the MCL1 inhibitors from Servier, Amgen or Astra-Zeneca. Of the authors want to use the other cells lines they use to test BCL-2 and/or BCL-XL dependencies, they should first check the literature to see whether this has been done or perform the experiment themselves.

I am still unconvinced by the authors' interpretation of the cellular results presented in Figure 5c,d,e. The A549 cells are already resistant to ABT199 and even more so to S55746 (starting IC50 value of 1 μ M and 74 μ M). This is well above the values that would be expected for a sensitive cell line (as demonstrated by the data in RS4,11 in Figure 5b). Another, and very likely, interpretation of the data in Figure 5e is that the peptide has off-target activity which doesn't depend on BCL-2.

Regarding on-target mode of action: since the CPs are weakly active, this will be a challenging task indeed. The most efficient strategy is to use a panel of cell lines with defined dependency on the various pro-survival proteins (RS4,11 is one of them for BCL-2 dependency). This question remains open and will only be resolved with more cell penetrant and/or more active peptides (as the authors pointed out).

In conclusion: the biophysical data (up to Figure 4) is reasonably strong and interesting. The cellular data remains questionable.

Minor remarks:

P19, line 437: "It is known that small-molecule inhibitors targeting BCL-2/BCL-XL require a high affinity to induce conformational changes of the hydrophobic surface groove": this is incorrect. There are compounds with weak affinity that induce large conformational change. Examples can be found in early analogues in the development of ABT-737 or the BCL-XL selective compounds WEHI-539. The P2 pocket in particular is flexible and accommodate small molecules with weaker affinity (single digit μ M).

The authors have now included SD values for all results, which is great. I would like to remind them that these SDs should have the same number of significant figures as the mean (i.e. line 347 page 15: 0.04 +/- 0.003 should be rounded at 0.04 +/- 0.01).

I have noted that all the sensorgrams in the supp figures show only the fitted curves but not the raw data + fitted curves. This should be corrected.

Fig 5d: please do not truncate bar graphs.

Reviewer #2 (Remarks to the Author):

I am mostly satisfied with the revised manuscript that the authors have submitted. However, there were still a few points that have not been addressed sufficiently.

* a new validation report for 7Y90 has not been submitted. I am still concerned about the Rfree being so high for a high resolution dataset especially when all the statistics appear to be normal.

* the data completeness in the validation reports does not match that quoted in Table 1.

* Line 147-148: in the added text the authors describe an eight-fold change as "dramatically" lower - this is an overstatement of the significance and should be amended.

* Line 178-185: I find this discussion on SASAs unconvincing. Is it not possible that the smaller SASAs in the cp1-bound structures are reflective of the lower affinity? The BH3 peptides used as comparisons are in the low nM range, and venetoclax is sub/single digit nM. It is true that cp1 doesn't induce conformational change of $\alpha 3$ and $\alpha 4$, but it is also significantly less potent than venetoclax and ABT-737 for its target protein so the direct comparison seems a bit of stretch.

Reviewer #4 (Remarks to the Author):

Authors have fully addressed all the comments from the initial review of the manuscript. I congratulate them for their excellent research work.

Again, we thank all the reviewers for their careful evaluation and constructive guidance in improving our manuscript.

REVIEWER COMMENTS

Reviewer #1 (Remarks to the Author):

Overall, the authors have addressed a number of the concerns raised. In particular they have added repeats and a few critical experiments, especially in cells that are dependent on BCL-2 for survival, helping greatly with benchmarking the activity of these designed cyclic peptides vs small molecules.

There are however some remaining concerns that the additional data has not fully addressed:

The authors seem to have misinterpreted my point about the need to include data for RS4,11 cells: it wasn't to validate ABT-199's or ABT-263's on target activity, which is well documented. It was to benchmark the cyclic peptides in a validated cell line that is indeed known to be BCL-2 dependent. The result of this experiment are clearly indicating that the CPs do not efficiently induce BCL-2-dependent cell death and that the death observed at these high concentration, in all cell lines, is probably non-specific. Also, levels of pro-survival proteins do not dictate their dependence to these proteins. Dependencies is analysed using validated tool compounds such as ABT-199, A-1331852 or the MCL1 inhibitors from Servier, Amgen or Astra-Zeneca. Of the authors want to use the other cells lines they use to test BCL-2 and/or BCL-XL dependencies, they should first check the literature to see whether this has been done or perform the experiment themselves.

We fully understand the reviewer's point on testing the BCL-2 and/or BCL-XL dependencies on specific and proper cell lines that have been validated by the known tool compounds. For small-molecule inhibitors, this should be a very effective and straightforward method, further supplemented by stable knockout or overexpressing cell lines for each target to explore the on-target effects or the mutational effects. However, a key issue has to be emphasized here is that the cellular administration of peptides, especially for the CPs or others aiming at intracellular targets, is largely limited by their membrane permeability. The early stabled peptides targeting BCL-2 and the latest CPs targeting intracellular proteins both support this notion (*Science* 2004, 305:1466-1470; *Cell* 2022, 185: 3950-3965.e25). Therefore, we may need to first optimize our CPs in the future to improve their membrane-penetrating efficiency and selectivity. Otherwise, even if we choose a variety of BCL-2 pathway-dependent cell lines to test, it would be very difficult to achieve a clear conclusion due to the variations of CP penetration for different cell lines.

I am still unconvinced by the authors' interpretation of the cellular results presented in

Figure 5c,d,e. The A549 cells are already resistant to ABT199 and even more so to S55746 (starting IC50 value of 1 μ M and 74 μ M). This is well above the values that would be expected for a sensitive cell line (as demonstrated by the data in RS4,11 in Figure 5b). Another, and very likely, interpretation of the data in Figure 5e is that the peptide has off-target activity which doesn't depend on BCL-2.

Thank you for your comments. Actually, we first attempted to construct RS4:11 cells that stably overexpress BCL-2 WT or G101V proteins, but failed despite multiple attempts. Thus, we selected the BCL-2 low-expressing A549 as a tool cell line. Indeed, this cell line is not BCL-2 highly dependent as the RS4:11. However, to a certain extent, this system could demonstrate that the activities of ABT-199 and S55746 are influenced by the G101V mutation as reported, and those of the CPs are relatively less affected by the same mutation, potentially rendered by their special mode of action. In addition, we could not fully rule out the possibility that some off-target activities of CPs are also involved as the reviewer pointed out. Corresponding interpretation has been added at the end of the Results section (Lines 446-448 on Page 19).

Regarding on-target mode of action: since the CPs are weakly active, this will be a challenging task indeed. The most efficient strategy is to use a panel of cell lines with defined dependency on the various pro-survival proteins (RS4,11 is one of them for BCL-2 dependency). This question remains open and will only be resolved with more cell penetrant and/or more active peptides (as the authors pointed out).

We appreciate the reviewer's recognition of the intrinsic limitations of our CPs upon validating their on-target effects in cells. We will attempt to improve the membrane penetration efficiency and activity of these CPs. Meanwhile in this work, we added a new cellular experiment on the cytochrome C release by mitochondria, to test whether the relatively weak inhibitory activities of CPs are dependent on their on-target effects upon the apoptosis pathway (**Fig. 5f, g** and Lines 417-427 on Page 18).

In conclusion: the biophysical data (up to Figure 4) is reasonably strong and interesting. The cellular data remains questionable.

Many thanks for the positive comment on our biophysical data, as well as the criticism on our cellular data. In this revision, we modified certain conclusions from the cell-based experiments (highlighted in blue in the text) and added a new assay to test the potential on-target effects of CPs (**Fig. 5f, g**).

Minor remarks:

P19, line 437: "It is known that small-molecule inhibitors targeting BCL-2/BCL-XL require a high affinity to 438 induce conformational changes of the hydrophobic surface

groove”: this is incorrect. There are compounds with weak affinity that induce large conformational change. Examples can be found in early analogues in the development of ABT-737 or the BCL-XL selective compounds WEHI-539. The P2 pocket in particular is flexible and accommodate small molecules with weaker affinity (single digit μM).

Thank you for pointing this out. We have modified the corresponding description in the manuscript (Lines 451-453 on Page 19).

The authors have now included SD values for all results, which is great. I would like to remind them that these SDs should have the same number of significant figures as the mean (i.e. line 347 page 15: 0.04 ± 0.003 should be rounded at 0.04 ± 0.01).

Many thanks for the suggestion. This value has been modified accordingly (**Table 1** and Lines 347-348 on Page 15).

I have noted that all the sensorgrams in the supp figures show only the fitted curves but not the raw data + fitted curves. This should be corrected.

According to this suggestion and the data requirements of the journal, we have provided all the raw data for the SPR experiments in an .XLSX format in the “Source data” folder.

Fig 6d: please do not truncate bar graphs.

This figure has been redrawn following the reviewer’s comment.

Reviewer #2 (Remarks to the Author):

I am mostly satisfied with the revised manuscript that the authors have submitted. However, there were still a few points that have not been addressed sufficiently.

* a new validation report for 7Y90 has not been submitted. I am still concerned about the R_{free} being so high for a high resolution dataset especially when all the statistics appear to be normal. the data completeness in the validation reports does not match that quoted in Table 1.

Many thanks for the comments. A new validation report for 7Y90 along with the original data files (output.mtz, final.mtz, final.pdb, and scale.log) have been provided this time for further examination by the reviewer. For this structure, the quality of its diffraction pattern was relatively poor, with some diffraction points being split. Although the density of cp1 was fitted well, there were still noise signals in the density map, especially for a flexible loop region with 16 residues

that lack electron density. These factors may have contributed to the high R-free value. The resolution of this data set has been reduced to 2.1 Å, with an outer shell Linear R-fac 0.448. After refinement, the final R-factor/R-free values have become relatively reasonable at 0.220/0.275. We have also updated the statistics in Supplementary Table 2.

* Line 147-148: in the added text the authors describe an eight-fold change as "dramatically" lower - this is an overstatement of the significance and should be amended.

According to this suggestion, we have modified the corresponding description in the manuscript (Lines 147 to 149 on Page 7).

* Line 178-185: I find this discussion on SASAs unconvincing. Is it not possible that the smaller SASAs in the cp1-bound structures are reflective of the lower affinity? The BH3 peptides used as comparisons are in the low nM range, and venetoclax is sub/single digit nM. It is true that cp1 doesn't induce conformational change of $\alpha 3$ and $\alpha 4$, but it is also significantly less potent than venetoclax and ABT-737 for its target protein so the direct comparison seems a bit of stretch.

Many thanks for the constructive comments. In the first round of our revision, Reviewer #4 advised us to quantitatively compare the differences across the binding pockets of different complexes. The results of SASA calculation indeed support that cp1 binding does not cause conformational changes of $\alpha 3$ and $\alpha 4$. However, these SASA values may not correlate directly and positively with the affinities of binders, as we only included certain key residues on the binding surface (as illustrated in **Supplementary Fig. 7**). Nevertheless, as pointed out by Reviewer #1, the unique mode of action of CPs, which does not insert into the pocket, may be a key factor contributing to its relatively lower affinity. The corresponding descriptions have been modified within the manuscript (Lines 175-181 on Page 8). Also, we have added description of the algorithm used for SASA calculation in the Methods section (Lines 679-681 on Page 34).

Reviewer #4 (Remarks to the Author):

Authors have fully addressed all the comments from the initial review of the manuscript. I congratulate them for their excellent research work.

We sincerely appreciate your helpful suggestions.

REVIEWER COMMENTS

Reviewer #1 (Remarks to the Author):

The authors have addressed my questions regarding the SPR experiments. The additional data file was great.

With regards to the cellular activity, the authors were not able to robustly demonstrate on target activity in this assays. While I am convinced they have been able to develop relatively selective binders, further optimisation would be required to see they activity translate in cells. The cytochrome c release assay is in theory a good addition. However: 1) It doesn't remove the issue of cell permeability of the peptides, unless the experiment is conducted with permeabilised cells (as described in the literature) or with isolated mitochondria.

2) the control peptide BIP-V5, is to my knowledge not a validated control BAX inhibitor.

3) this experiment is classically conducted with level of cyt c in membrane vs supernatant. As it stands this experiment doesn't provide any additional information regarding the on target activity of the compounds.

Minor remark about the new paragraph describing the cytoplasmic c data:

- "These results implied that CPs may inhibit Jurkat cells...": I am not sure what the authors refer to when they write "may inhibit Jurkat cells"?

- "...by interfering the BCL-2 family-dependent mitochondrial apoptosis pathway.": should be "...by interfering with the BCL-2 family-dependent mitochondrial apoptosis pathway."

Reviewer #2 (Remarks to the Author):

Thank you for sending the data files and the new validation report for 7Y90. I am now happy with this dataset.

I have one remaining issue, however. The dataset completeness values in the validation reports does not match that reported in Supplementary Table 2 for any of the structures. The largest discrepancy is 7YA5, which is reported as having an overall completeness in the table as 92.1%, but the validation report states the overall completeness as only 77.6%. Have the right files been uploaded to the PDB? This discrepancy should be addressed.

We sincerely thank the reviewers for their constructive comments and suggestions on our manuscript. Corresponding modifications have been highlighted in blue in the text.

Reviewer #1 (Remarks to the Author):

The authors have addressed my questions regarding the SPR experiments. The additional data file was great.

With regards to the cellular activity, the authors were not able to robustly demonstrate on target activity in this assays. While I am convinced they have been able to develop relatively selective binders, further optimisation would be required to see they activity translate in cells. The cytochrome c release assay is in theory a good addition. However: 1) It doesn't remove the issue of cell permeability of the peptides, unless the experiment is conducted with permeabilised cells (as described in the literature) or with isolated mitochondria.

Many thanks for the above comments. We agree with the reviewer that further optimization on our CPs is necessary to translate their activities into cells, with the cell permeability being one of the most critical limiting factors. We have modified the corresponding descriptions in the text, to make it clear to the readers that the membrane penetration of the CPs and their clinical relevance remain to be established (Lines 410 to 413 on Page 18).

2) the control peptide BIP-V5, is to my knowledge not a validated control BAX inhibitor.

Although BIP-V5 has not been proven to be a universal BAX inhibitor, it has been shown as a BAX-mediated apoptosis inhibitor through inhibiting the expression of BAX in certain cell lines (e.g. STF-cMyc cells and H9c2 cells, *Diabetes*. 2007 May;56(5):1259-67; *European Journal of Pharmacology* 769 (2015) 257–265). In our experiment, we first treated Jurkat cells with BIP-V5 at different concentrations and for different duration time. We found that after treatment by BIP-V5 at 50 μ M for 24 h, the expression of BAX proteins obviously decreased (Fig. 5f). We have also revised the relevant description in our manuscript (Lines 417 to 420 on Page 18).

3) this experiment is classically conducted with level of cyt c in membrane vs supernatant. As it stands

this experiment doesn't provide any additional information regarding the on target activity of the compounds.

For the cyt c release experiment, indeed the classical conduction is to detect both the release of cyt c in mitochondria and its increase in supernatant. However, in our experiment we found it was very difficult to detect cyt c in supernatant. We infer the possible reason could be that the CPs need a relative longer treatment time during which the cyt c in supernatant tends to be degraded. This experiment may only serve as an additional piece of evidence for the activity of our CPs. The related description has been modified in the manuscript (Lines 415 to 426 on Page 18).

Minor remark about the new paragraph describing the cytoplasmic c data:

- "These results implied that CPs may inhibit Jurkat cells...": I am not sure what the authors refer to when they write "may inhibit Jurkat cells"?

Thank you for pointing out this vague description. This sentence has been rephrased in the manuscript (Lines 424 to 426 on Page 18).

- "...by interfering the BCL-2 family-dependent mitochondrial apoptosis pathway.": should be "...by interfering with the BCL-2 family-dependent mitochondrial apoptosis pathway."

Accordingly, this sentence has been modified in the manuscript (Lines 424 to 426 on Page 18).

Reviewer #2 (Remarks to the Author):

Thank you for sending the data files and the new validation report for 7Y90. I am now happy with this dataset.

I have one remaining issue, however. The dataset completeness values in the validation reports does not match that reported in Supplementary Table 2 for any of the structures. The largest discrepancy is 7YA5, which is reported as having an overall completeness in the table as 92.1%, but the validation report states the overall completeness as only 77.6%. Have the right files been uploaded to the PDB? This discrepancy should be addressed.

We greatly appreciate the reviewer's careful examination. In Supplementary Table 2, the data completeness of 7YA5 is presented based on the raw data statistics in the scale.log file from the HKL3000 with the resolution range of 50-1.85 Å (Figure R1a). However, the completeness shown in validation in validation report is based on the value after refinement with the resolution range of 29.22-1.85 Å (Figure R1b). This could be the reason for the discrepancy of total average completeness in Supplementary Table 2 and in validation report. Normally, the completeness values should be similar before and after refinement, just like our other 5 datasets (with a high symmetry and space group of C222₁). However, the space group of 7YA5 is P2₁, and unfortunately the X-ray data was collected with the high-resolution shells (2.0-1.85 Å) showing only a low completeness (Figure R1a). All the datasets and structures reported in this work have been released by RCSB PDB. The statistics in our Supplementary Table 2 are shown as same as the values reported in the PDB.

Figure R1. Statistics about data completeness. a. Statistics showing % data completeness in raw data scal.log file. **b.** Data and refinement statistics in the validation report of 7YA5.

REVIEWERS' COMMENTS

Reviewer #2 (Remarks to the Author):

The authors have addressed all my concerns.

Reviewer #2 (Remarks to the Author):

The authors have addressed all my concerns.

We sincerely thank the reviewer for constructive suggestions helping improve our paper.